# Paradoxical dominant negative activity of an immunodeficiency-associated activating *PIK3R1* variant

**Patsy R Tomlinson[1,2], Rachel G Knox[1,2,3], Olga Perisic[4], Helen Su[5], Gemma V Brierley[1,6], Roger L Williams[4], Robert K Semple[1,3,7,8]\***

[1]The University of Cambridge Metabolic Research Laboratories, Wellcome Trust-MRC Institute of Metabolic Science, Cambridge, United Kingdom; [2]MRC Metabolic Diseases Unit, Wellcome Trust-MRC Institute of Metabolic Science, Cambridge, United Kingdom; [3]The National Institute for Health Research Cambridge Biomedical Research Centre, Cambridge, United Kingdom; [4]MRC Laboratory of Molecular Biology, Cambridge, United Kingdom; [5]Laboratory of Clinical Immunology & Microbiology, Intramural Research Program, National Institute of Allergy and Infectious Disease, National Institutes of Health, Bethesda, United States; [6]Department of Comparative Biomedical Science, The Royal Veterinary College, London, United Kingdom; [7]Centre for Cardiovascular Science, University of Edinburgh, Edinburgh, United Kingdom; [8]MRC Human Genetics Unit, Institute of Genetics and Cancer, University of Edinburgh, Edinburgh, United Kingdom

**\*For correspondence:**
rsemple@exseed.ed.ac.uk

## eLife Assessment

This **important** study reports on PIK3R1 mutations and a paradoxical mechanism of PIK3R1 signaling. The strength of evidence for the study is mostly **convincing**, as conclusions are supported by a variety of mutational strategies and cellular systems to look at interactions among signaling pathways.

**Abstract** *PIK3R1* encodes three regulatory subunits of class IA phosphoinositide 3-kinase (PI3K), each associating with any of three catalytic subunits, namely p110α, p110β, or p110δ. Constitutional *PIK3R1* mutations cause diseases with a genotype-phenotype relationship not yet fully explained: heterozygous loss-of-function mutations cause SHORT syndrome, featuring insulin resistance and short stature attributed to reduced p110α function, while heterozygous activating mutations cause immunodeficiency, attributed to p110δ activation and known as APDS2. Surprisingly, APDS2 patients do not show features of p110α hyperactivation, but do commonly have SHORT syndrome-like features, suggesting p110α hypofunction. We sought to investigate this. In dermal fibroblasts from an APDS2 patient, we found no increased PI3K signalling, with p110δ expression markedly reduced. In preadipocytes, the APDS2 variant was potently dominant negative, associating with Irs1 and Irs2 but failing to heterodimerise with p110α. This attenuation of p110α signalling by a p110δ-activating PIK3R1 variant potentially explains co-incidence of gain-of-function and loss-of-function *PIK3R1* phenotypes.

## Introduction

*PIK3R1* gene products (p85α, p55α, and p50α) are essential for class IA phosphoinositide 3-kinase (PI3K) signalling. Each of the three protein products can bind any of three catalytic subunits, p110α, β, and δ, encoded by *PIK3CA*, *PIK3CB*, and *PIK3CD,* respectively. This stabilises and inhibits the catalytic subunits in the basal state and confers responsiveness to upstream stimuli including receptor tyrosine kinase activation, enabled by two phosphotyrosine-binding SH2 domains shared by all PIK3R1 products (*Fruman et al., 2017*).

PI3K signalling mediates responses to a myriad of cues, including growth factors, hormones such as insulin, and processed antigens, and so recent discovery that genetic perturbation of PIK3R1 in humans disrupts growth, insulin action, and immunity is no surprise. Beyond this headline observation lies considerable complexity, however, and important questions about genotype-phenotype correlations in PIK3R1-related disorders are unresolved.

Some PIK3R1 mutations reduce basal inhibition of catalytic subunits, usually due to disruption of the inhibitory inter-SH2 domain, and are found in cancers (*Philp et al., 2001*) and vascular malformations with overgrowth (*Cottrell et al., 2021*). In both diseases, hyperactivated PI3Kα, composed of heterodimers of *PIK3R1* products and p110α, drives pathological growth. Distinct inter-SH2 domain *PIK3R1* mutations, mostly causing skipping of exon 11 and deletion of residues 434–475, hyperactivate PI3Kδ in immune cells, causing highly penetrant monogenic immunodeficiency (*Deau et al., 2014*; *Lucas et al., 2014b*). This phenocopies the immunodeficiency caused by genetic activation of p110δ itself, which is named <u>a</u>ctivated <u>P</u>I3K <u>d</u>elta <u>s</u>yndrome 1 (APDS1) (*Angulo et al., 2013*; *Lucas et al., 2014a*). The PIK3R1-related syndrome, discovered shortly afterwards, is thus named APDS2.

Despite ubiquitous *PIK3R1* expression, features attributable to PI3Kα activation have been described in only a single case of APDS2 to date (*Wentink et al., 2017*). In that case, the causal variant (N564K) was a constitutional activating mutation associated with cancer rather than one of the common APDS2 variants. Biochemical studies have suggested that apparently selective p110δ activation occurs because APDS2 mutations derepress p110δ, the predominant immune system catalytic subunit, more than p110α, due to differences in the inhibitory contacts between *PIK3R1* products and different catalytic subunits (*Dornan et al., 2017*).

More surprisingly, phenotypic overlap is reported between APDS2 and SHORT syndrome. SHORT syndrome, named for the characteristic developmental features (<u>s</u>hort stature, <u>h</u>yperextensibility, <u>h</u>ernia, <u>o</u>cular depression, <u>R</u>ieger anomaly, and <u>t</u>eething delay), is caused by loss of PI3Kα function due to disruption of the phosphotyrosine-binding C-terminal SH2 domain (*Chudasama et al., 2013*; *Dyment et al., 2013*; *Thauvin-Robinet et al., 2013*). Like APDS2 it is stereotyped and highly penetrant. It features short stature, insulin resistance, and dysmorphic features (*Avila et al., 2016*). In recent years, both individual case reports (*Bravo García-Morato et al., 2017*; *Petrovski et al., 2016*; *Ramirez et al., 2020*; *Szczawińska-Popłonyk et al., 2022*) and larger case series *Elkaim et al., 2016*; *Jamee et al., 2020*; *Maccari et al., 2023*; *Nguyen et al., 2023*; *Olbrich et al., 2016*; *Petrovski et al., 2016* have established that many people with APDS2 have overt features of SHORT syndrome, while, more generally, linear growth impairment is common in APDS2, but not in APDS1. These clinical observations are bolstered by the impaired linear growth and increased in utero mortality reported in mice with knock-in of the common causal APDS2 mutation in Pik3r1 (*Nguyen et al., 2023*). All people with SHORT-APDS2 overlap syndromes have well-established activating PIK3R1 mutations in the inter-SH2 domain implicated in APDS2, but none have characteristic SHORT synfdrome mutations, which are usually in the C-terminal SH2 domain. Conversely, no patient with a classical SHORT syndrome mutation has been shown to have immunodeficiency. There is thus now convincing evidence of a syndrome with features of both gain and loss of PI3K function. The mechanistic basis of this is unexplained.

PI3K activity is determined not just by activity of individual PI3K holoenzymes, but also by subunit stoichiometry. Regulatory and catalytic subunits stabilise each other upon binding (*Brachmann et al., 2005*; *Yu et al., 1998*), and only regulatory subunit monomers are stable enough to be detected in mammalian cells (*Yu et al., 1998*). Molar excess of regulatory subunits over catalytic subunits gives rise to free regulatory subunits that compete with holoenzyme for phosphotyrosine binding (*Ueki et al., 2002*). This has been invoked to explain increased insulin sensitivity on knockout or reduction of p85α alone (*Mauvais-Jarvis et al., 2002*), of p55α and p50α (*Chen et al., 2004*) or of all *Pik3r1* products (*Fruman et al., 2000*). The 'free p85' model has been contested by some reports suggesting no excess p85 (*Geering et al., 2007*; *Zhao et al., 2006*), however most observations, including some

made recently using in vivo tagging and pulldown of PI3K components (*Tsolakos et al., 2018*), suggest that monomeric p85α can be seen in vivo in some tissues.

We now address mechanisms underlying human mixed gain- and loss-of-function phenotypes observed in association with PIK3R1 mutations using primary cells, cell models, and in vitro enzyme assays and suggest that they are explained by competing effects of APDS2 PIK3R1 mutations on PI3K activity and stability.

## Results

### PI3K subunit expression and signalling in APDS2 fibroblasts

Disorders in the *PIK3CA*-related overgrowth spectrum (PROS) feature soft tissue overgrowth and are caused by heterozygous, mosaic activating mutations in *PIK3CA*, encoding the p110α catalytic subunit of PI3Kα (*Madsen et al., 2018*). In this group, basal hyperactivation of PI3Kα signalling can be easily discerned in dermal fibroblasts from affected areas of the body (*Lindhurst et al., 2012*). *PIK3R1*, like *PIK3CA*, is ubiquitously expressed, yet overgrowth is not reported in APDS2 caused by heterozygous, constitutional activating mutations in *PIK3R1*. Dermal fibroblasts strongly express *PIK3R1*, but no studies to date have evaluated whether PI3K activity is increased in APDS2.

We began by assessing dermal fibroblasts cultured from a previously described 32-year-old Turkish woman with APDS2 due to the common causal PIK3R1 mutation. This affects a splice donor site that causes skipping of exon 11, leading to in-frame deletion of 42 amino acids (434–475 inclusive) in the inter-SH2 domain, which is shared by all PIK3R1 isoforms (Patient A.1 in *Lucas et al., 2014b*; *Figure 1—figure supplement 1*). These cells were compared to cells from healthy control volunteers, or from people with PROS. Confirmation of expression of all pathogenic mutations was undertaken by cDNA sequencing prior to further study (*Figure 1—figure supplement 2A*). We found that truncated p85α was expressed, but at a much lower protein level than full-length wild-type (WT) p85α (*Figure 1A*, *Figure 1—figure supplement 2B and C*). Despite this, no increase in basal or insulin-stimulated AKT phosphorylation was seen in APDS2 cells compared to cells from WT volunteers or from people with PROS and activating *PIK3CA* mutations H1047L or H1047R (*Figure 1A–C*, *Figure 1—figure supplement 3A and B*). Although insulin-induced AKT phosphorylation was lower in fibroblasts from the one APDS2 patient studied compared to controls, we have previously reported extensive variability in insulin-responsiveness of primary dermal fibroblasts. Moreover even primary cells from a patient expressing high levels of the SHORT syndrome-associated p85α Y657X did not show attenuated insulin action (*Huang-Doran et al., 2016*). The reduced insulin action in APDS2 cells in the current study thus should not be overinterpreted until reproduced in further APDS2 cells.

Previous studies have suggested that the truncated, APDS2-causal p85α variant exerts a much greater activating effect on p110δ, which has a more restricted tissue expression, including immune cells, than p110α, which is ubiquitous (*Dornan et al., 2017*). We found that both p110α and p110δ were easily detectable in control dermal fibroblasts, however p110δ was almost absent in APDS2 fibroblasts compared to controls, with lower levels also seen in PROS cells (*Figure 1A and D*, *Figure 1—figure supplement 3D*). p110α was unchanged in APDS2 cells but modestly increased in PROS cells (*Figure 1A and E*, *Figure 1—figure supplement 3C*). Collectively these findings suggest that any ability of the APDS2 PIK3R1 variant in skin cells to activate PI3K may be overcome by reduced protein levels of p110δ, likely through reduced binding and/or reduced stabilisation of p110δ by the mutant regulatory subunit.

### Overexpressed PIK3R1 ΔExon11 is potently dominant negative in 3T3-L1 preadipocytes

We next turned to a well-established cellular system allowing conditional overexpression of *PK3R1* alleles of interest. We have previously shown that overexpression of two SHORT syndrome PIK3R1 variants – R649W and Y657X – impair insulin signalling and adipocyte differentiation of murine 3T3-L1 preadipocytes, consistent with impaired PI3Kα function and with the lipodystrophy and insulin resistance seen in SHORT syndrome (*Huang-Doran et al., 2016*). To assess the effect of the APDS2 ΔEx11 in the same, non-immunological, cell context, we used lentiviral vectors, as previously described (*Huang-Doran et al., 2016*), to generate clonal 3T3-L1 cell lines allowing conditional, tuneable overexpression of PIK3R1 variants in response to doxycycline (*Figure 2—figure supplement 1*). For signalling

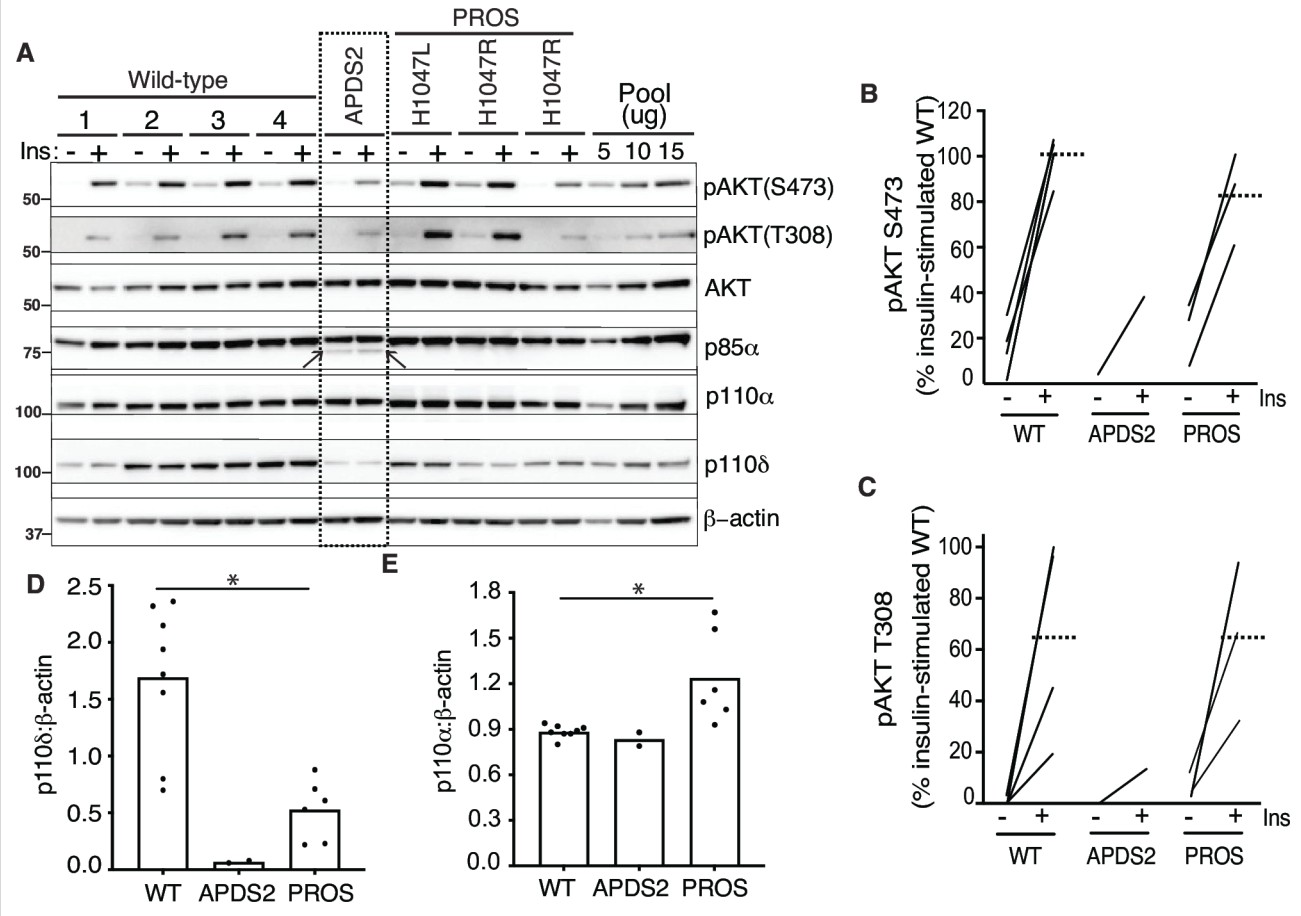

**Figure 1.** Phosphoinositide 3-kinase (PI3K) subunit expression and signalling in primary dermal fibroblasts. Immunoblotting of AKT, AKT phosphorylated at threonine 308 (T308) or serine 473 (S473), p85α, p110δ, and p110α and are shown with and without stimulation by 100 nM insulin (Ins) for 10 min. β-Actin is shown as a loading control, with different amounts of pooled lysate (Pool) used to demonstrate signal intensity in the linear range. Molecular weight markers (in kDa) are indicated to the left. Results are shown from four healthy controls (wild-type [WT]; 1–4), one patient with activating p110 delta syndrome 2 (APDS2) due to the p85α Δexon11 variant, and three patients with PIK3CA-related overgrowth spectrum (PROS) caused by the activating *PIK3CA* mutations indicated. (**A**) Immunoblots, with the truncated p85α Δexon11 variant arrowed. (**B–E**) Quantification of immunoblot bands from three independent experiments are shown for phosphoAKT-S473, phosphoAKT-T308, p110δ, and p110α, respectively. Each point represents data from one of the patient cell lines in the immunoblots. Paired datapoints ± insulin are shown in (**B**) and (**C**), and dotted lines mark means. Asterisks indicate a significant difference. More detailed statistical analysis including 95% confidence intervals for the paired mean differences for these comparisons are shown in *Figure 1—figure supplement 2*.

The online version of this article includes the following source data and figure supplement(s) for figure 1:

**Source data 1.** Original gel image files for western blot analysis displayed in *Figure 1A* (and shown in higher magnification in *Figure 1—figure supplement 2B and C*), including images of two further experimental replicates included in analysis.

**Source data 2.** PDF file containing original western blots for *Figure 1A* (shown in higher magnification in *Figure 1—figure supplement 2B and C*), indicating excerpts displayed in figures and replicates included in analysis.

**Figure supplement 1.** Schematic illustrating the PIK3R1 variants studied.

**Figure supplement 2.** Further characterisation of primary dermal fibroblasts studied.

**Figure supplement 3.** Full statistical analysis of data presented in main *Figure 1* analysis of insulin-induced increase in (**A**) AKT S473/4 and (**B**) T308/9 phosphorylation.

studies, we also generated cells conditionally overexpressing the PROS- and cancer-associated PIK3CA H1047R mutation as a positive control for increased PI3Kα signalling, while for differentiation studies we used previously reported lines conditionally expressing PIK3R1 R649W or Y657X (*Huang-Doran et al., 2016*; *Figure 1—figure supplement 1*). In undifferentiated cells, we first confirmed doxycycline-dependent overexpression of p85α or p110α transgenes (*Figure 2A*), before assessing

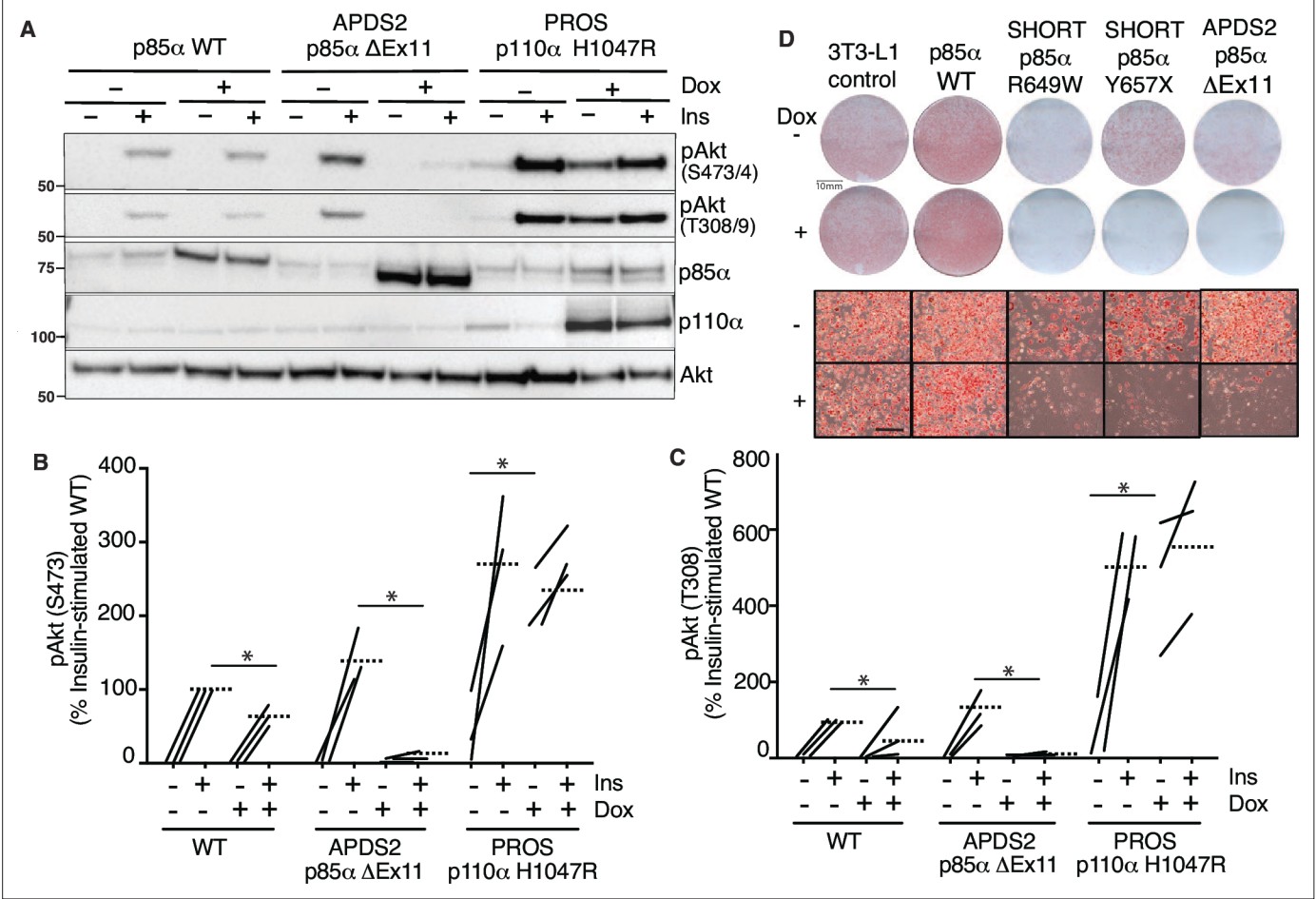

**Figure 2.** Blunted insulin signalling in 3T3-L1 preadipocyte models of activating p110 delta syndrome 2 (APDS2) and SHORT syndrome. Immunoblotting of Akt, Akt phosphorylated at threonine 308 (T308) or serine 473 (S473), p85α, and p110α and are shown with and without stimulation with 100 nM insulin (Ins) for 10 min. Molecular weight markers (in kDa) are indicated to the left. Cells were treated with doxycycline (Dox) 1 µg/mL for 72 hr prior to insulin stimulation as indicated. (**A**) One immunoblot representing three experiments is shown. (**B, C**) Quantification of immunoblot bands from all three independent experiments shown for phosphoAkt-S473 and phosphoAkt-T308, respectively. Paired datapoints ± insulin are shown, and dotted lines mark means. Asterisks indicate a significant difference. More detailed statistical analysis including 95% confidence intervals for the paired mean differences for these comparisons are shown in *Figure 2—figure supplement 2*. (**D**) Staining for neutral lipid with Oil Red O of 3T3-L1 cells at day 10 of adipocyte differentiation. Induction of transgene expression by 1 µg/mL Dox throughout differentiation is shown. Images of entire plates are shown above, with representative bright-field microscopy images below. Scale bars on micrographs are 100 µm.

The online version of this article includes the following source data and figure supplement(s) for figure 2:

**Source data 1.** Original gel image files for western blot analysis displayed in *Figure 2A*, including images of two further experimental replicates included in analysis.

**Source data 2.** PDF file containing original western blots for *Figure 2A*, indicating excerpts displayed in figures and replicates included in analysis.

**Source data 3.** Original microscopic and macroscopic images of Oil Red O-stained 3T3-L1 adipocytes, including images presented in *Figure 2D* and further replicates.

**Figure supplement 1.** Schematic illustrating experimental design for 3T3-L1 studies.

**Figure supplement 2.** Full statistical analysis of data presented in main *Figure 2*.

**Figure supplement 3.** The effect of graded expression of wild-type or disease-associated p85α on 3T3-L1 preadipocytes.

**Figure supplement 3—source data 1.** Original gel image files for western blot analysis displayed in *Figure 2—figure supplement 3*, including images of one further experimental replicate.

**Figure supplement 3—source data 2.** PDF file containing original western blots for *Figure 2—figure supplement 3*, indicating excerpts displayed in figure and replicate.

basal and insulin-stimulated Akt phosphorylation. As expected, overexpression of oncogenic H1047R p110α strongly increased basal Akt phosphorylation (*Figure 2A–C*, *Figure 2—figure supplement 2A and C*) with no additional increase on insulin stimulation (*Figure 2A–C*, *Figure 2—figure supplement 2B and D*). Surprisingly, however, not only did overexpression of the APDS2 ΔEx11 p85α not increase basal PI3Kα signalling, but it also potently inhibited insulin-induced Akt phosphorylation, consistent with a strong dominant negative action on pathway activation (*Figure 2A–C*, *Figure 2—figure supplement 2*). Overexpressing WT p85α mildly reduced insulin-stimulated Akt phosphorylation (*Figure 2A–C*, *Figure 2—figure supplement 2B and C*), although WT p85α was not expressed to as high a level as ΔEx11 p85a. In keeping with impaired PI3Kα activity, overexpression of the ΔEx11 variant also severely impaired adipocyte differentiation, assessed by triglyceride accumulation in response to a standard differentiation protocol (*Figure 2D*). A similar effect was seen on overexpressing SHORT syndrome variants R649W and Y657X (*Figure 2D*).

To assess whether the inhibitory effect of ΔEx11 p85α might be an artefact caused by strong overexpression, doxycycline titration was used to assess whether a low level of overexpression might unmask signalling hyperactivation. However, no such hyperactivation was seen, and instead, a graded diminution of insulin-induced AKT phosphorylation was observed from the lowest to highest levels of p85a ΔEx11 overexpression (*Figure 2—figure supplement 3*).

## Effect of PIK3R1 mutations on PI3K activity in vitro

Given this evidence that APDS2-associated PIK3R1 ΔEx11 potently inhibits PI3Kα signalling when overexpressed in 3T3-L1 preadipocytes, we next sought to investigate the biochemical basis of this observation. First, we assessed the effect of disease-causing PIK3R1 mutations on basal and phosphotyrosine-stimulated activity of purified PI3Kα, β, and δ holoenzyme in a previously described reconstituted in vitro system (*Burke et al., 2012*). As well as the APDS2 ΔEx11 mutation, we selected three SHORT syndrome-associated mutations to study. These were the most common causal mutation, R649W, which abolishes phosphotyrosine binding by the C-terminal SH2 (cSH2) domain (*Chudasama et al., 2013*), Y657X, which truncates the cSH2 domain (*Huang-Doran et al., 2016*; *Kwok et al., 2020*), and E489K which, atypically, lies in the inter-SH2 domain where most cancer, overgrowth, and APDS2-associated mutations lie (*Thauvin-Robinet et al., 2013*). PIK3R1 E489K-containing primary cells were previously suggested to show basal hyperactivation (*Thauvin-Robinet et al., 2013*).

Wild-type and SHORT syndrome mutant holoenzymes were successfully purified for in vitro assay, but despite multiple attempts, ΔEx11 holoenzyme could only be made in small amounts under identical conditions, and moreover was unstable on storage, precluding further study. Such instability of in vitro synthesised ΔEx11 holoenzyme was previously reported (*Dornan et al., 2017*). Also in keeping with previous reports (*Chudasama et al., 2013*; *Dornan et al., 2017*; *Dornan et al., 2020*), p85α R649W showed severely reduced phosphotyrosine-stimulated activity in complex with p110α, with highly significant but lesser loss of function seen for Y657X and E489K (*Figure 3A*, *Figure 3—figure supplement 1A*). No increase in basal activity was seen for any variant.

We also assessed whether any of the SHORT syndrome variants affect function of PI3Kβ and found that all variants impaired phosphotyrosine-stimulated activity of PI3Kβ. Again, this impairment was less for p85α Y657X than for R649W, and in this case only very mild for E489K (*Figure 3B*, *Figure 3—figure supplement 1B*). For PI3Kδ, p85α R649W conferred severe loss of function, as for other isoforms (*Figure 3C*, *Figure 3—figure supplement 1C*), suggesting that the absence of immunodeficiency in SHORT syndrome is not accounted for by selective inhibition of PI3Kα function by causal mutations.

## Binding of p110α by mutant p85α

Given the potent dominant negativity of APDS2-related p85α ΔEx11 in cells, and the instability of p85α ΔEx11-containing PI3K holoenzyme in vitro, we next used immunoprecipitation to assess binding of p110α by this and other p85α variants in cells. p85α was easily detected in anti-p110α immunoprecipitates from all cell lines at baseline (*Figure 4*). No increase in p110α levels was seen on conditional overexpression of WT or R649W p85α. Given the known instability of monomeric p110α, this suggests that all p110α is bound to p85α before overexpression. Although WT endogenous p85α may have been replaced by heterologously overexpressed p85α in these cells, this could not be detected without a size difference of the variant from WT. For cells overexpressing the SHORT syndrome-associated p85α Y657X, the truncated variant was strongly co-immunoprecipitated, accounting for nearly all of

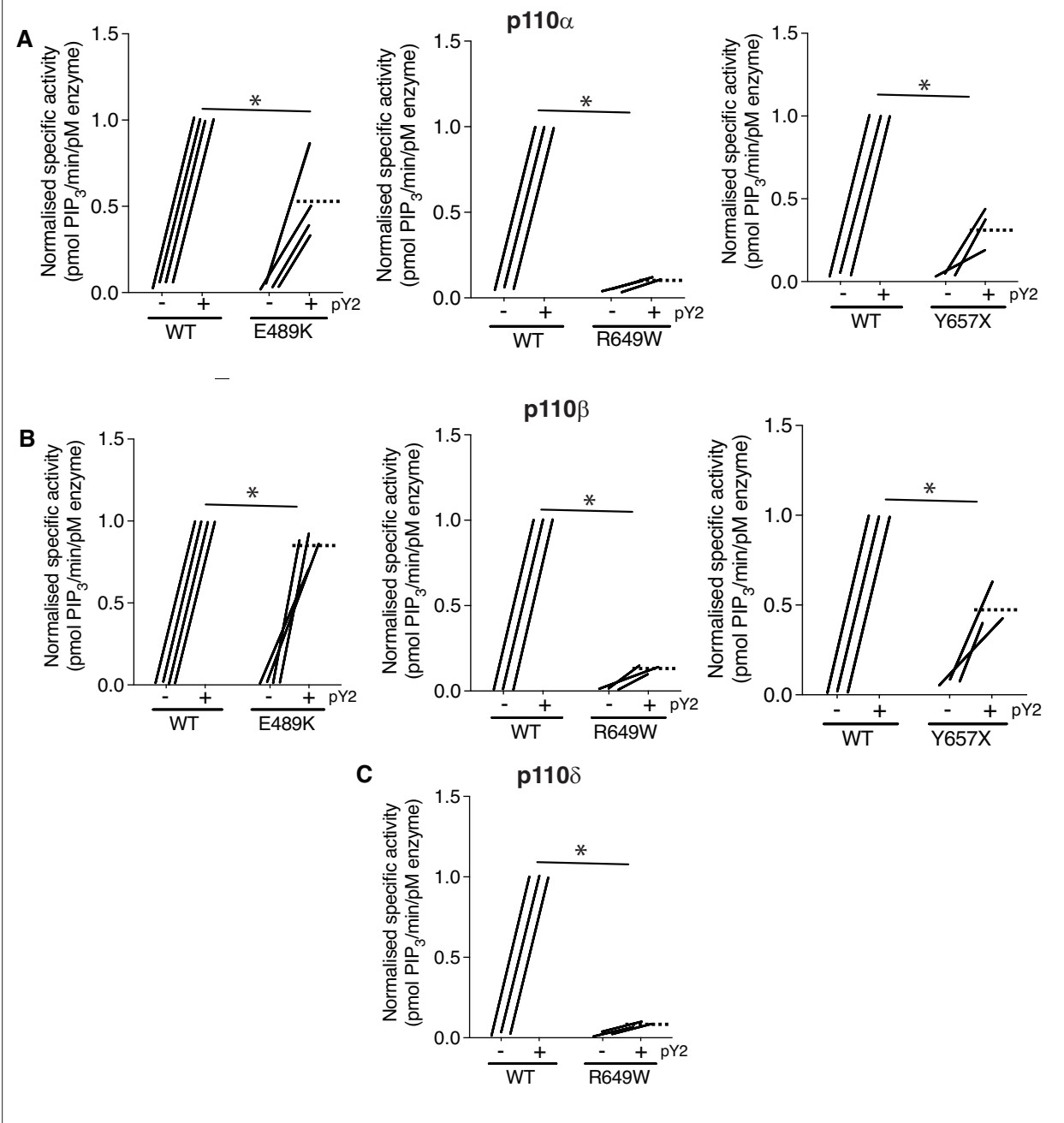

**Figure 3.** SHORT syndrome p85α mutations impair phosphotyrosine-stimulated phosphoinositide 3-kinase (PI3K) activity. Lipid kinase activity of purified recombinant PI3K complexes generated using baculoviral expression in *Sf9* cells was measured using a modified fluorescence polarisation assay. Wild-type (WT) p85α or p85α SHORT syndrome mutations, E489K, R649W, or Y657X bound to either (**A**) p110α, (**B**) p110β, or (**C**) p110δ were assayed for basal and bisphosphotyrosine (pY2)-stimulated lipid kinase activity. Dotted lines mark means, and asterisks indicate a significant difference between the bisphosphotyrosine (pY2)-stimulated state for WT and comparator mutant p85α. More detailed statistical analysis including 95% confidence intervals for the paired mean differences for these comparisons are shown in *Figure 3—figure supplement 1*.

The online version of this article includes the following figure supplement(s) for figure 3:

**Figure supplement 1.** Full statistical analysis of data presented in main *Figure 3*.

the p85α signal in anti-p110α immunoprecipitate. This demonstrates preserved binding of p110α by mutant p85α (*Figure 4*). In sharp contrast, although truncated p85α ΔEx11 was easily detected in cell lysates before immunoprecipitation and in supernatant after immunoprecipitation (arrowed in *Figure 4*), no truncated p85α ΔEx11 was seen in p110α immunoprecipitates, and no change in p110α

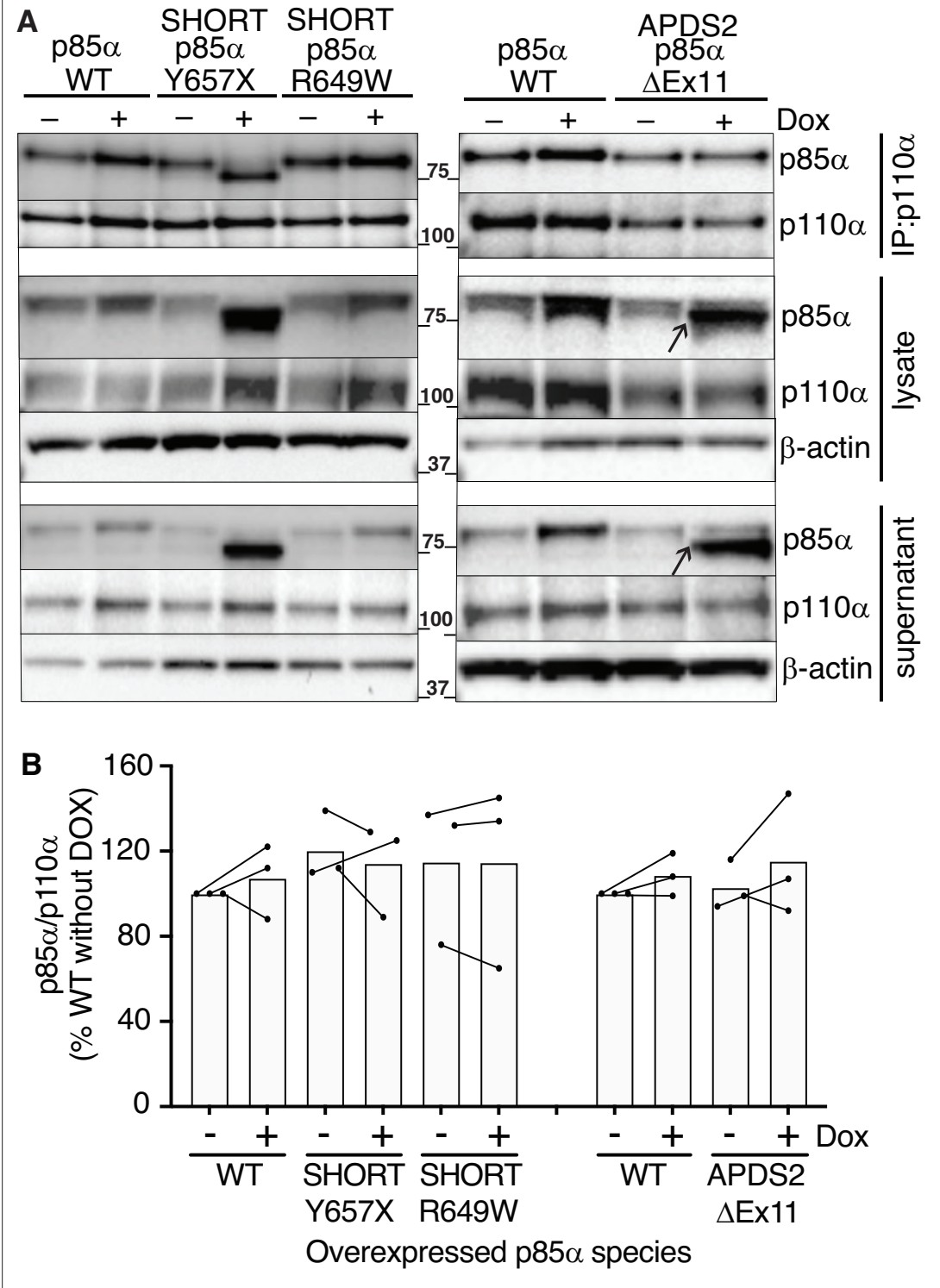

**Figure 4.** Ability of pathogenic p85α variants to bind p110α, assessed by co-immunoprecipitation. Results of immunoblotting of anti-p110α immunoprecipitates from 3T3-L1 cells expressing wild-type (WT), activating p110 delta syndrome 2 (APDS2)-associated or SHORT syndrome-associated mutant p85α under the control of doxycycline (Dox) are shown. (**A**) One representative immunoblot of immunoprecipitate, cell lysate prior to immunoprecipitation, and post immunoprecipitation supernatant is shown. Molecular weight markers (in kDa) are indicated to the left. between gel images. (**B**) Quantification of immunoblot bands from immunoprecipitates from three independent experiments, expressed as a percentage relative to the intensity of the band in WT cells without Dox exposure. Co-immunoprecipitated p85α is shown normalised to immunoprecipitated p110α from

*Figure 4 continued*

all three independent experiments. Datapoints from the same experiment ± Dox are connected by lines. No significant differences were found among conditions.

The online version of this article includes the following source data for figure 4:

**Source data 1.** Original gel image files for western blot analysis displayed in *Figure 4A*, including images of two further experimental replicates included in analysis.

**Source data 2.** PDF file containing original western blots for *Figure 4A*, indicating excerpts displayed in figures and replicates included in analysis.

expression was detected (*Figure 4*). This suggests that this truncated APDS2 causal variant does not supplant endogenous, full-length p85α binding to p110α, despite overexpression. This argues against destabilisation of p110α as the mechanism of the observed dominant negative activity.

## Effect of PIK3R1 mutations on insulin-induced PI3K recruitment to IRS1/2

As APDS2 p85α ΔEx11 does not appear to displace WT p85α from p110α, despite strong overexpression, it is likely that there are high levels of truncated p85α unbound to p110α in the cell. This may be important, as p85α mediates recruitment of PI3K to activated tyrosine kinase receptors and their tyrosine-phosphorylated substrates, including the insulin-receptor substrate proteins Irs1 (*Myers et al., 1992*) and Irs2 (*Sun et al., 1995*). Excess free regulatory subunits compete with heterodimeric PI3K holoenzyme for binding to these phosphotyrosines (*Ueki et al., 2002*), raising the possibility that excess free, truncated APDS2 p85α ΔEx11 may exert its inhibitory action similarly by outcompeting PI3K holoenzyme for phosphotyrosine binding.

To assess this possibility, we again used the 3T3-L1 cellular model to determine whether overexpression of disease-causing p85α variants impairs recruitment of p110α to Irs1 and Irs2. Irs1 was immunoprecipitated with or without conditional p85α overexpression and with or without insulin stimulation. Overexpression of WT p85α had no effect on basal or insulin-induced p110α recruitment to Irs1 (*Figure 5A and B*, *Figure 5—figure supplement 1*). In contrast, overexpression of either p85α R649W or Y657X sharply reduced insulin-stimulated p110α recruitment to Irs1 on insulin stimulation (*Figure 5A and B*, *Figure 5—figure supplement 1*). In keeping with the inability of the R649W cSH2 domain to bind phosphotyrosines, p85α R649W was also not recruited to Irs1, while overexpression of Y657X increased stimulated but not basal p85α recruitment (*Figure 5A and C*, *Figure 5—figure supplement 2*). This suggests a pure defect in PI3K holoenzyme recruitment for R649W. For Y657X the signalling defect may have mixed mechanisms, with reduced activation by phosphotyrosine seen in in vitro studies coupled to increased abundance of monomeric mutant p85α, leading to recruitment of non p110α-bound mutant p85α to Irs1.

In keeping with our finding of severely attenuated insulin signalling upon p85a ΔEx11 overexpression (*Figure 2*), overexpression of this mutant p85α abolished p110α recruitment to Irs1 (*Figure 5A and B*, *Figure 5—figure supplement 1*). However, non-p110α-bound p85a ΔEx11 was strongly recruited to Irs1 even in the absence of insulin stimulation (*Figure 5A and C*, *Figure 5—figure supplement 2*). This suggests that although p85a ΔEx11 does not effectively compete with WT p85α for binding to p110α, it has preserved or possibly enhanced ability to bind Irs1. This gives the mutant p85α properties that render it a more potent endogenous inhibitor of PI3K signalling than free WT p85α. Experiments conducted with immunoprecipitation of Irs2 instead of Irs1 yielded closely similar findings for all p85α species (*Figure 6* and *Figure 6—figure supplements 1 and 2*).

## Discussion

Murine genetic studies of Pik3r1 have proved complicated to interpret, due in part to functional redundancy among PI3K regulatory subunits, and in part to unanticipated effects of perturbing PI3K subunit stoichiometry. This has left several important questions about in vivo functions of Pik3r1 unresolved. Identification of a series of human genetic disorders caused by constitutional PIK3R1 mutations over the past 10 years has given fresh impetus to the field and has been mechanistically illuminating.

Homozygous truncating PIK3R1 mutations abolishing p85α expression while preserving p55α and p50α produce agammaglobulinaemia (*Conley et al., 2012*; *Tang et al., 2018*; *Figure 1—figure*

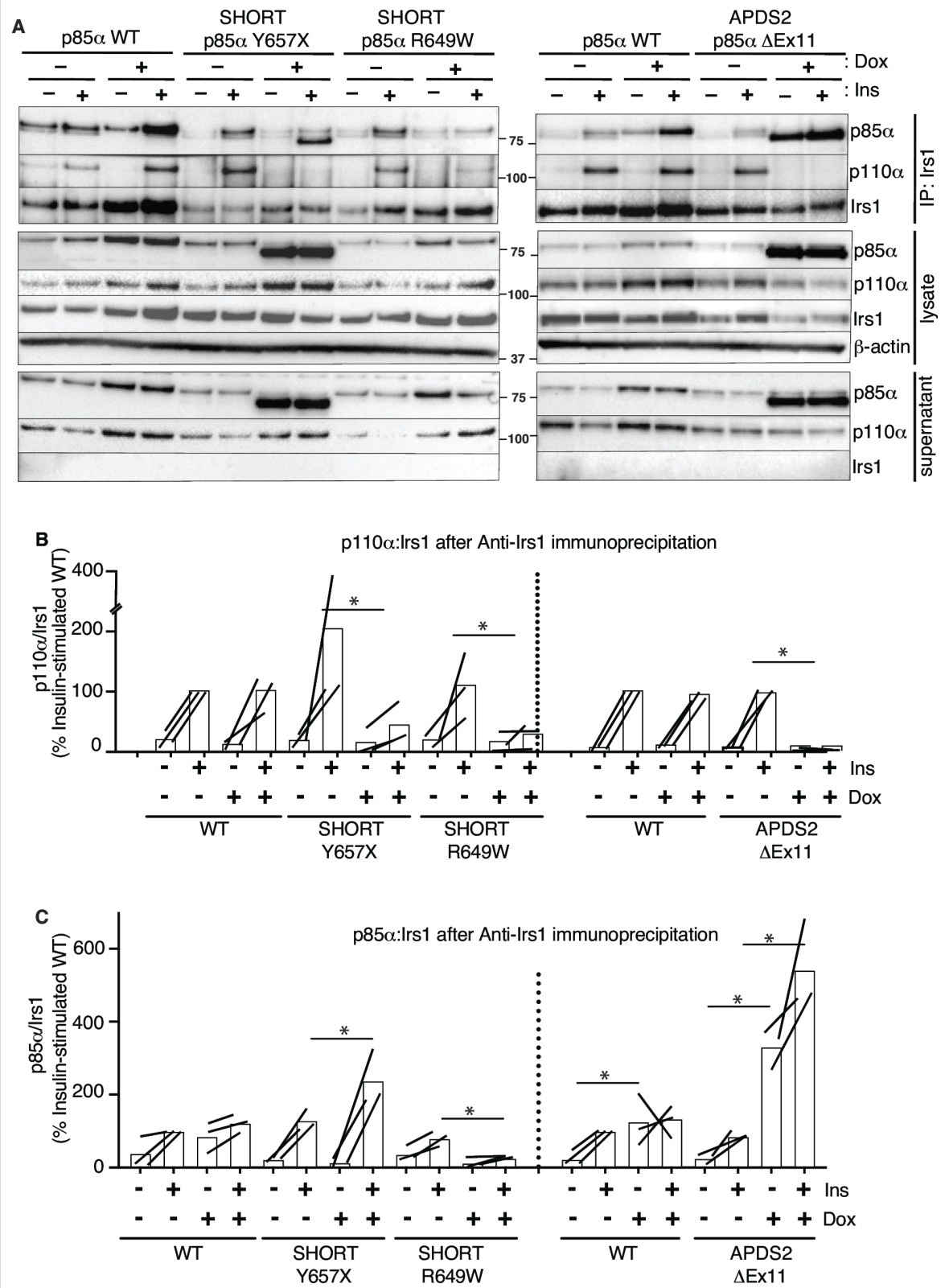

**Figure 5.** Attenuated insulin-induced association of p110α with Irs1 in the presence of activating p110 delta syndrome 2 (APDS2) and SHORT syndrome mutant p85α. Results of immunoblotting of anti-Irs1 immunoprecipitates from 3T3-L1 cells expressing wild-type, APDS2-associated, or SHORT syndrome-associated mutant p85α under the control of doxycycline (Dox) are shown. Treatment with 100 nM insulin (Ins) is indicated. (**A**) One representative immunoblot of immunoprecipitate, cell lysate prior to immunoprecipitation, and post immunoprecipitation supernatant is shown. Two

*Figure 5 continued on next page*

*Figure 5 continued*

separate sets of gels, including independent wild-type controls, are shown on left and right. Molecular weight markers (in kDa) are indicated between gel images. (B, C) Quantification of immunoblot bands from immunoprecipitates from three independent experiments. Immunoprecipitated p110α is shown normalised to immunoprecipitated Irs1 from all three independent experiments in (B), and immunoprecipitated p85α similarly in (C). Datapoints from the same experiment -± insulin are connected by lines. Asterisks indicate significant differences induced by transgene overexpression (i.e. plus versus minus doxycycline). More detailed statistical analysis including 95% confidence intervals for the paired mean differences for these comparisons are shown in *Figure 5—figure supplements 1 and 2*.

The online version of this article includes the following source data and figure supplement(s) for figure 5:

**Source data 1.** Original gel image files for western blot analysis displayed in *Figure 5A*, including images of two further experimental replicates included in analysis.

**Source data 2.** PDF file containing original western blots for *Figure 5A*, indicating excerpts displayed in figures and replicates included in analysis.

**Figure supplement 1.** Full statistical analysis of data presented in main *Figure 5A and B*.

**Figure supplement 2.** Full statistical analysis of data presented in main *Figure 5A and C*.

---

supplement 1). This resembles the immunodeficiency reported in *Pik3r1* knockout mice (*Fruman et al., 1999*; *Suzuki et al., 1999*) and suggests an essential, non-redundant function of p85α in B cell development in humans. Thereafter, human genetics has provided more novel insights. *PIK3R1* mutations were identified in SHORT syndrome in 2013 (*Chudasama et al., 2013*; *Dyment et al., 2013*; *Thauvin-Robinet et al., 2013*), nearly all in the C-terminal SH2 domain, which together with the N-terminal SH2 domain enables PI3K recruitment to activated RTKs (*Rordorf-Nikolic et al., 1995*). SHORT syndrome features short stature and insulin resistance consistent with impaired ligand-induced p110α action, a phenotype distinct from the enhanced insulin sensitivity produced by genetic ablation of one or more *Pik3r1* products in mice (*Chen et al., 2004*; *Fruman et al., 2000*; *Mauvais-Jarvis et al., 2002*). Mice with SHORT syndrome mutations knocked in were generated after human findings, and faithfully reproduce the human phenotype (*Kwok et al., 2020*; *Solheim et al., 2018*; *Winnay et al., 2016*), confirming that expression of a signalling-impaired PIK3R1 has different consequences to Pik3r1 knockout. No immunodeficiency has been described associated with classic SHORT syndrome mutations, however, suggesting a selective effect on PI3Kα, but the basis of this selectivity has not previously been investigated.

In contrast to SHORT syndrome, mutations in the inter-SH2 domain of PIK3R1, mostly leading to skipping of exon 11, were shown in 2014 to activate PI3K in vitro and to cause immunodeficiency (APDS2) (*Deau et al., 2014*; *Lucas et al., 2014b*) similar to that caused by activating mutations in p110δ (APDS1) (*Angulo et al., 2013*; *Lucas et al., 2014a*). Neither overgrowth nor metabolic features of APDS2 have been described to suggest p110α hyperactivation, and indeed short stature is common in APDS2 (*Elkaim et al., 2016*; *Jamee et al., 2020*; *Maccari et al., 2023*; *Olbrich et al., 2016*; *Petrovski et al., 2016*), with a growing number of APDS2 patients described with features of SHORT syndrome (*Bravo García-Morato et al., 2017*; *Maccari et al., 2023*; *Nguyen et al., 2023*; *Petrovski et al., 2016*; *Ramirez et al., 2020*; *Szczawińska-Popłonyk et al., 2022*). Moreover mice with the common APDS2 causal Pik3r1 variant knocked in show impaired growth and in utero survival, unlike APDS2 murine models (*Nguyen et al., 2023*).

Thus, study of distinct PIK3R1-related syndromes shows that established loss-of-function PIK3R1 mutations produce phenotypes attributable to selectively impaired PI3Kα hypofunction, while activating mutations produce phenotypes attributable to selectively increased PI3Kδ signalling. Indeed, not only do such activating mutations not produce phenotypes attributable to PI3Kα activation, but they surprisingly have features characteristic of impaired PI3Kα function.

Lack of overgrowth in APDS2 has been attributed to greater ability of APDS2 PIK3R1 variants to activate p110δ than p110α (*Dornan et al., 2017*), but the co-occurence of gain- and loss-of-function phenotypes has not been explained to date. Our findings suggest that the explanation may lie in competing effects of APDS2 PIK3R1 variants to activate PI3Kδ on one hand, as previously shown (*Angulo et al., 2013*; *Dornan et al., 2017*; *Lucas et al., 2014b*), while on the other hand interfering with PI3Kα through the dominant negative effect of non-p110α-bound mutant p85α.

The low expression of truncated p85a ΔEx11 we described in dermal fibroblasts is similar to observations made in lymphocytes; however, in lymphocytes this is associated with increased basal AKT phosphorylation (*Deau et al., 2014*; *Lucas et al., 2014b*) that is abolished by p110δ inhibition (*Lucas*

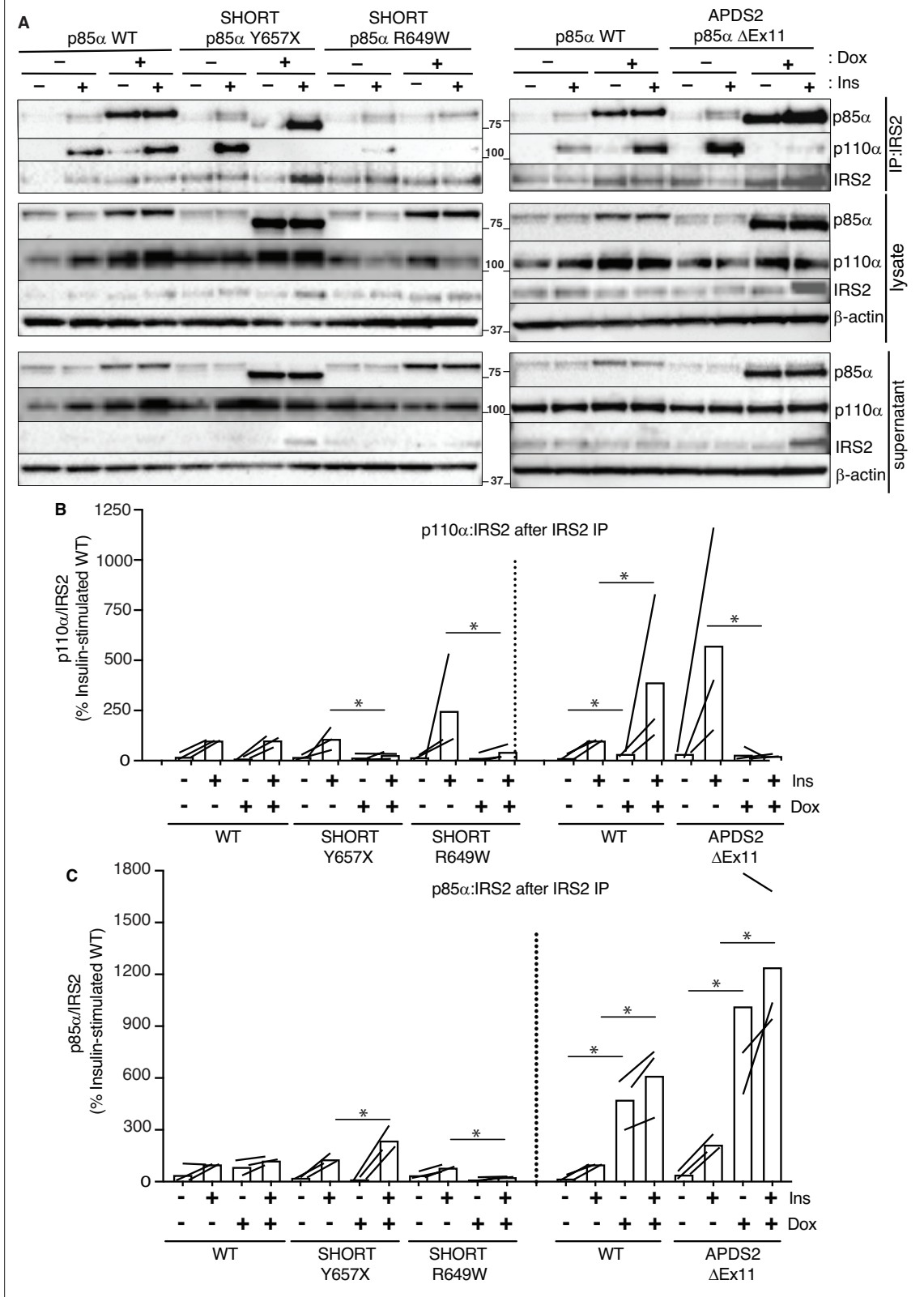

**Figure 6.** Attenuated insulin-induced association of p110α with Irs2 in the presence of activating p110 delta syndrome 2 (APDS2) and SHORT syndrome mutant p85α. Results of immunoblotting of anti-Irs2 immunoprecipitates from 3T3-L1 cells expressing wild-type, APDS2-associated, or SHORT syndrome-associated mutant p85α under the control of doxycycline (Dox) are shown. Treatment with 100 nM insulin (Ins) is indicated. (**A**) One representative immunoblot of immunoprecipitate, cell lysate prior to immunoprecipitation, and post immunoprecipitation supernatant is shown. Two

*Figure 6 continued on next page*

*Figure 6 continued*

separate sets of gels, including independent wild-type controls are shown on left and right. Molecular weight markers (in kDa) are indicated between gel images. (**B, C**) Quantification of immunoblot bands from immunoprecipitates from three independent experiments. Immunoprecipitated p110α is shown normalised to immunoprecipitated Irs2 from all three independent experiments in (**B**), and immunoprecipitated p85α similarly in (**C**). Datapoints from the same experiment ± insulin are connected by lines. More detailed statistical analysis including 95% confidence intervals for the paired mean differences for these comparisons are shown in *Figure 6—figure supplements 1 and 2*.

The online version of this article includes the following source data and figure supplement(s) for figure 6:

**Source data 1.** Original gel image files for western blot analysis displayed in *Figure 6A*, including images of two further experimental replicates included in analysis.

**Source data 2.** PDF file containing original western blots for *Figure 6A*, indicating excerpts displayed in figures and replicates included in analysis.

**Figure supplement 1.** Full statistical analysis of data presented in *Figure 6A and B*.

**Figure supplement 2.** Full statistical analysis of data presented in *Figure 6A and C*.

*et al., 2014b*). P110δ is also expressed in fibroblasts, but protein levels were reduced in the cells studied, consistent with the previously reported observation that when expressed in insect cells the PI3K holoenzyme containing p85a ΔEx11 has a low yield and is unstable (*Dornan et al., 2017*). We speculate that in cells with low endogenous p110δ protein expression, the destabilising effect of mutant PIK3R1 predominates over its activating effect.

The balance between expression and signalling in different cells may be a fine one, however, as transient overexpression of FLAG-tagged p85a ΔEx11 did increase AKT phosphorylation in 3T3 fibroblasts in a previous study, although expression of p110α, β, and δ was not determined (*Deau et al., 2014*). We find, in contrast, that overexpression of untagged p85a ΔEx11 has a strong inhibitory effect on insulin signalling, and we were unable to identify a window of overexpression where increased AKT phosphorylation could be observed. We further demonstrated that most or all of the mutant p85α expressed is not bound to p110α, while unbound mutant p85α still binds to Irs1/2, effectively competing with PI3K holoenzyme. The observation that p85a ΔEx11 can associate with Irs1/2 is in agreement with reports that p85α ΔEx11-containing PI3Kα and PI3Kδ can be stimulated by pY2-peptides (*Dornan et al., 2017*; *Dornan et al., 2020*) and that p85a ΔEx11 is recruited to tyrosine-phosphorylated LAT in T cells (*Lucas et al., 2014b*). The competition we suggest between unbound mutant p85α and PI3K holoenzyme for binding to the activated RTKs is in keeping with longstanding evidence that free p85 downregulates PI3K signalling through the same competitive mechanism (*Thorpe et al., 2017*; *Ueki et al., 2002*), with a single study suggesting in addition that overexpression of tagged p85α leads to Irs1 sequestration with free p85α in cytosolic foci where PI(3,4,5)P3 production does not occur (*Luo et al., 2005*).

The current study has limitations. We have studied primary cells from only a single APDS2 patient, and in the 3T3-L1 cell model, we did not determine whether p110δ protein could be detected. If not, this could explain the lack of detectable AKT phosphorylation with induction of Pik3r1 ΔEx11. Indeed, previous pharmacological studies in 3T3-L1 adipocytes has shown that selective inhibition of p110δ or p110β does not alter insulin-induced phosphorylation of any protein studied in the PI3-K pathway, attesting to the dominance of p110α in insulin action in this cell model (*Knight et al., 2006*). Our study moreover raises further questions. Full-length p85α subunits can homodimerise (*Cheung et al., 2015*; *Harpur et al., 1999*; *LoPiccolo et al., 2015*), and it is speculated that homodimers may outcompete p85α/p110 heterodimers for binding to activated Irs1 due to configuration of the four SH2 domains. In the 3T3-L1 preadipocyte APDS2 models, p85a ΔEx11 expression was high despite impaired p110α association, but whether this is composed of monomeric or homodimeric p85α, and whether mutant p85α expression leads to sequestration of Irs1 remote from the insulin receptor, as previously suggested for tagged WT p85α (*Luo et al., 2005*), is undetermined.

In summary, it is already established that: (1) genetic activation of PIK3CD causes immunodeficiency without disordered growth, while (2) inhibition of PIK3R1 recruitment to RTKs and their substrates impairs growth and insulin action, without immunodeficiency, despite all catalytic subunits being affected and (3) loss of p85α alone causes immunodeficiency. The current study, coupled with prior reports, suggests that the common APDS2 mutation in PIK3R1 has mixed consequences, producing greater hyperactivation of p110δ than p110α, based on subtle differences in the inhibitory interactions of regulatory and catalytic subunits, while also destabilising PI3K holoenzyme and exerting

dominant negative activity on WT PIK3R1 function. We suggest that these competing activating and inhibitory consequences are finely balanced, potentially differing among tissues, leading to mixed clinical profiles of gain- and loss-of-function features.

# Materials and methods

## Key resources table

| Reagent type (species) or resource | Designation | Source or reference | Identifiers | Additional information |
|---|---|---|---|---|
| Gene (*Homo sapiens*) | PIK3R1 | GenBank UniProt | NM_181523.3 P27986.1 | p85α protein product only studied |
| Cell line (*H. sapiens*) | Dermal fibroblasts Wild-type 1; 2; 3; 4 | https://doi.org/10.1172/jci.insight.88766; https://doi.org/10.1038/ng.2332 | | Described in Ethics |
| Cell line (*H. sapiens*) | Dermal fibroblasts APDS2 | https://doi.org/10.1084/jem.20141759 | | Described in Ethics |
| Cell line (*H. sapiens*) | Dermal fibroblasts PROS H1047L; H1047R; H1047R | This study: https://doi.org/10.1038/ng.2332 | | Described in Ethics |
| Cell line (*Spodoptera frugiperda*) | Sf9 cells | Thermo Fisher | #11496015 | |
| Cell line (*Mus musculus*) | 3T3-L1 preadipocytes | Zenbio | Lot 3T3062104 | Passage 8 |
| Cell line (*Escherichia coli*) | Stellar competent cells | Takara | #636763 | |
| Cell line (*E. coli*) | MAX Efficiency DH10Bac Competent Cells | Invitrogen | #10361–012 | |
| Recombinant DNA reagent (*H. sapiens*) | Hsp85a_pACEBac1 | https://doi.org/10.1016/j.str.2011.06.003 | | Generated in Williams Lab, MRC-LMB; expressing Human p85α (UniProtKB P27986.1) |
| Recombinant DNA reagent (*H. sapiens*) | Hsp85a-Y657*_pACEBac1 | Human p85α-Y657* | This study | Based on Hsp85a_pACEBac1; Described in Baculovirus generation section of Materials and methods |
| Recombinant DNA reagent (*H. sapiens*) | Hsp85a-R649W_pACEBac1 | Human p85α-R649W | This study | Based on Hsp85a_pACEBac1; Described in Baculovirus generation section of Materials and methods |
| Recombinant DNA reagent (*H. sapiens*) | Hsp85a-E489K_pACEBac1 | Human p85α-E489K | This study | Based on Hsp85a_pACEBac1 |
| Recombinant DNA reagent (*H. sapiens*) | Hsp85a-dEx11_pACEBac1 | Human p85α-dEx11 | This study | Based on Hsp85a_pACEBac1; Described in Baculovirus generation section of Materials and methods |
| Recombinant DNA reagent (*H. sapiens*) | Hsp110a_pFastBacHT B | https://doi.org/10.1016/j.str.2011.06.003 | | Generated in Williams Lab, MRC-LMB; expressing Human p110α (UniProtKB P42336.2) |
| Recombinant DNA reagent (*H. sapiens*) | Hsp110b_pACEBac1 | https://doi.org/10.1016/j.str.2011.06.003 | | Generated in Williams Lab, MRC-LMB; expressing Human p110β (UniProtKB P42338.1) |
| Recombinant DNA reagent (*H. sapiens*) | Hsp110d_pFastBacHT B | https://doi.org/10.1016/j.str.2011.06.003 | | Generated in Williams Lab, MRC-LMB; expressing Human p110δ (UniProtKB O00329.2) |
| Recombinant DNA reagent | pEN_Tmcs | Addgene | RRID:Addgene_25751 | |
| Recombinant DNA reagent | pSLIK-Hygro | Addgene | RRID:Addgene_25737 | |
| Recombinant DNA reagent (*H. sapiens*) | Hsp85a-dEx11_pSLIK-Hygro | Human p85α-dEx11 | This study | Generation described in Generation of 3T3-L1 cells conditionally expressing p85α or p110α section of Materials and methods |
| Recombinant DNA reagent | pMDLg/pRRE | Addgene | RRID:Addgene_12251 | |

*Continued on next page*

*Continued*

| Reagent type (species) or resource | Designation | Source or reference | Identifiers | Additional information |
|---|---|---|---|---|
| Recombinant DNA reagent | pRSV-Rev | Addgene | RRID:Addgene_12253 | |
| Recombinant DNA reagent | pMD2.G | Addgene | RRID:Addgene_12259 | |
| Transfected construct (*H. sapiens* construct in *M. musculus* cells) | p85α WT; SHORT p85α Y657X; SHORT p85α R649W | https://doi.org/10.1172/jci.insight.88766 | | |
| Transfected construct (*H. sapiens* construct in *M. musculus* cells) | APDS2 p85α Δex11 | This study | | Described in Materials and methods |
| Transfected construct (*H. sapiens* construct in *M. musculus* cells) | PROS p110α H1047R | This study | | Described in Materials and methods |
| Antibody | Anti-p85a (Rabbit, monoclonal) | Cell Signaling Technology | RRID:AB_659889 | (Used at 1:1000 dilution) |
| Antibody | Anti-Phospho-AKT/Akt_T308 (Rabbit, polyclonal) | Cell Signaling Technology | RRID:AB_329828 | (Used at 1:1000 dilution) |
| Antibody | Anti-Phospho-AKT/Akt_S473 (Rabbit, polyclonal) | Cell Signaling Technology | RRID:AB_329825 | (Used at 1:1000 dilution) |
| Antibody | Anti-AKT/Akt (Mouse, monoclonal) | Cell Signaling Technology | RRID:AB_1147620 | (Used at 1:1000 dilution) |
| Antibody | Anti-p110a (Rabbit, monoclonal) | Cell Signaling Technology | RRID:AB_2165248 | (Used at 1:1000 and 1:50 dilution for immunoblotting and immunoprecipitation respectively) |
| Antibody | Anti-p110d (Rabbit, monoclonal) | Cell Signaling Technology | RRID:AB_2799043 | (Used at 1:1000 dilution) |
| Antibody | Anti-Irs1 (Rabbit, polyclonal) | Millipore | RRID:AB_2127890 | (Used at 1:500 dilution) |
| Antibody | Anti-Irs1 (Rabbit, polyclonal) | Cell Signaling Technology | RRID:AB_330333 | (Used at 1:1000 and 1:50 dilution for immunoblotting and immunoprecipitation respectively) |
| Antibody | Anti-Irs2 (Mouse, monoclonal) | Millipore | RRID:AB_11211231 | (Used at 1:1000 and 1:50 dilution for immunoblotting and immunoprecipitation respectively) |
| Antibody | Anti-β-actin (Rabbit, polyclonal) | Cell Signaling Technology | RRID:AB_330288 | (Used at 1:5000 dilution) |
| Antibody | HRP-linked Anti-rabbit IgG (Goat, polyclonal) | Cell Signaling Technology | RRID:AB_2099233 | (Used at 1:5000 dilution) |
| Antibody | HRP-linked Anti-mouse IgG (Horse, polyclonal) | Cell Signaling Technology | RRID:AB_330924 | (Used at 1:5000 dilution) |
| Peptide, recombinant protein | pY2 (PDGFRβ peptide 735- ESDGGYMDMSKDES-IDYVPMLDMKGDIKYADIE –767) | Cambridge Peptides | | |
| Commercial assay or kit | Quikchange II XL Site-Directed Mutagenesis Kit | Agilent | #200521 | |
| Commercial assay or kit | In-fusion Cloning Kit | Takara | #638909 | |
| Commercial assay or kit | Zymoclean Gel DNA Recovery Kit | Zymo Research | #D4001 | |
| Commercial assay or kit | Nucleospin Clean-up Kit | Takara | #740609.10 | |
| Commercial assay or kit | PI 3-Kinase Activity Fluorescence Polarisation Assay | Echelon Biosciences | #K-1100 | |

*Continued on next page*

*Continued*

| Reagent type (species) or resource | Designation | Source or reference | Identifiers | Additional information |
|---|---|---|---|---|
| Commercial assay or kit | Protein G Dynabeads | Life Technologies | #10003D | |
| Chemical compound, drug | Sphingomyelin; cholesterol; porcine brain phosphatidylcholine; phosphatidylethanolamine; phosphatidylserine; phosphoinositide-4,5-bisphosphate | Avanti Polar Lipids | | |

## Baculovirus generation

p85α point mutation expression constructs were created by site-directed mutagenesis of an Hsp85α_pACEBac1 plasmid using a QuikChange II XL Site-Directed Mutagenesis Kit (Agilent, 200521). In-Fusion cloning (Takara, 638909) was used to generate Hsp85α_pACEBac1_p85α ΔEx11. In brief, Hsp85α_pACEBac1 backbone was digested (BamHI/NotI) and purified using Zymoclean Gel DNA Recovery Kit (Zymo Research, D4001), while the cDNA insert was purified using NucleoSpin Clean-up Kit (Takara, 740609.10). Insert generation was performed by High-Fidelity PCR using primers designed to yield fragments containing exons 1–10 or 12–16 with the necessary overlap, with inserts verified by electrophoresis and purified using NucleoSpin Clean-up Kit. In-Fusion reactions were performed according to the manufacturer's guideline using 100 ng each of linearised plasmid and both inserts in 10 µL. Products were transformed into Stellar Competent cells (Takara, 636763). Purified plasmid inserts were sequenced and verified by restriction enzyme digest, exploiting loss of a DpnI site within excised exon 11.

Hsp85α_pACEBac1, Hsp110α_pACEBac1, Hsp110β_pACEBac1, and Hsp110δ_pFastBacHT B plasmids (*Burke et al., 2011*) encoding N terminally tandem His-tagged p110 subunits were used to generate bacmid DNA. MAX Efficiency DH10Bac Competent Cells (Invitrogen, 10361-012) were transformed with 40 ng plasmid DNA and turbid cultures plated onto agar containing 50 µg/mL kanamycin, 10 µg/mL gentamicin, 10 µg/mL tetracycline, 100 µg/mL X-Gal, and 40 µg/mL IPTG. Single colonies were picked, expanded, and purified at 2 days and bacmid concentrations quantified by NanoDrop 1000 Spectrophotometer (Thermo Scientific). 2–4 µg bacmid was transfected into Sf9 cells in six-well plates using Insect-XPRESS with L-Glutamine (Lonza, 12-730Q) and FuGENE HD (Promega E2311). Cells were incubated at 2°C for 5 days and bacmid YFP expression assessed by Leica DM IL LED Fluo microscope using a green fluorescent protein filter cube. Pooled virus from two wells of supernatant (P1 stock) was used to produce high titre P2 stock by transfection of $1.5 \times 10^6$ Sf9 cells/mL (Thermo Fisher, 11496015) in 450 mL using Insect-XPRESS with L-Glutamine and FuGENE HD. For catalytic subunits, 1–2 mL P2 stock was used for a further round of Sf9 transfection and expansion to generate P3 stock.

## Purification of PI3K holoenzymes

1.5–2 L of $1.5 \times 10^6$ Sf9 cells/mL were co-infected with 18 mL P3 catalytic subunit and 4 mL P2 regulatory subunit baculovirus. Non-infected Sf9 cells were negative controls, and a prior protein preparation of 320 kDa was the positive control. Cells were cultured at 27°C, harvested 48 hr post-infection, pelleted, washed, and stored at –80°C. Pellets were later lysed by sonication in buffer containing 20 mM Tris-HCl (pH 8.0), 300 mM NaCl, 20 mM imidazole (pH 8.0), and 1 mM TCEP (pH 7.0) at 4°C. Universal Nuclease (Thermo Fisher, 88702) was added to lysates before ultracentrifugation at 35,000 rpm for 35 min at 4°C. p85α/p110 heterodimers were pulled down via six tandem p110 N-terminal His tags, preventing purification of monomeric p85α, using tandem $Ni^{2+}$HisTrap Fast Flow columns (GE Healthcare) (equilibration buffer 20 mM Tris-HCl [pH 8.0], 100 mM NaCl, 20 mM imidazole, and 1 mM TCEP [pH 7.0]; elution buffer 20 mM Tris-HCl [pH 8.0], 100 mM NaCl, 200 mM imidazole, and 1 mM TCEP [pH 7.0]). Further purification utilised a heparin HiTrap Q HP column (GE Healthcare) equilibrated with 20 mM Tris-HCl (pH 8.0) and 1 mM TCEP (pH 7.0), with proteins eluted in 20 mM Tris-HCl (pH 8.0), 1 M NaCl, and 1 mM TCEP (pH 7.0). Eluted fractions were concentrated to ≥1 mg/mL using Millipore Amicon Ultra-15 Centrifugal Filter Units with Ultracel-50 membrane, and 1 mL concentrated fractions were gel-filtered on Superdex 200 16/60 columns (GE Healthcare)

equilibrated in 20 mM HEPES (pH 7.5), 100 mM NaCl, and 2 mM TCEP (pH 7.0). The p110 6-His tag was retained for proteins used in functional analyses. ÄKTA Protein Purification Systems (GE Healthcare) and UNICORN Control Software version 5.11 (GE Healthcare) were used for all purifications. Purity of eluted complexes was verified by SDS-PAGE, and purified proteins were quantified using a NanoDrop 1000 Spectrophotometer (Thermo Fisher). Protein concentration was calculated using the molecular extinction coefficient (assuming full cysteine reduction) determined by heterodimer sequence input to the ProtParam tool (ExPASy). Purified proteins were stored at –80°C in single-use aliquots.

Total injection volume onto gel filtration columns was kept at 1 mL. Yields of PI3K complexes were determined using the area under the curve (280 nm mAU per mL eluted protein) from gel filtration chromatograms, and normalised to the volume of Sf9 cells infected.

## Liposome preparation

Chloroform/methanol solutions of sphingomyelin, cholesterol, porcine brain phosphatidylcholine, porcine brain phosphatidylethanolamine, porcine brain phosphatidylserine, and porcine brain PI(4,5) P2 (Avanti Polar Lipids) were mixed to generate a preparation containing 5/10/15/45/20/5% (wt/vol) of each lipid, with a total lipid concentration of 5 mg/mL and final PI(4,5)P2 concentration 250 µg/ mL. Lipid preparations were dessicated under argon and then in a vacuum desiccator. Lipids were rehydrated in buffer containing 20 mM HEPES (pH 7.5), 100 mM KCl, and 1 mM EGTA (pH 8.0) using vortexing for 3 min, water bath sonication for 15 min, and 10 cycles between liquid nitrogen and a 43°C water bath. Unilamellar vesicles were generated by extrusion through polycarbonate filters with 100 nm pores, using a glass-tight syringe. Single-use aliquots were stored at –80°C.

## Fluorescence polarisation assay

PI(3,4,5)P3 production was measured by modified PI3-kinase activity fluorescence polarisation assay (Echelon Biosciences, Salt Lake City, UT, USA). 10 µL reactions in 384-well black microtitre plates used 1 mM liposomes containing 50 µM PI(4,5)P2, optimised concentrations of purified PI3K proteins, 100 µM ATP, 2 mM $MgCl_2$, with or without 1 µM tyrosine bisphosphorylated 33-mer peptide derived from mouse PDGFRβ residues 735–767, including phosphotyrosine at positions 740 and 751 ('pY2'; 735-ESDGG**Y**MDMSKDESID**Y**VPMLDMKGDIKYADIE-767; Cambridge peptides). Reactions were quenched with 5 µL of solution containing 20 mM HEPES (pH 7.5), 150 mM NaCl, 30 mM EDTA (pH 7.4), and 400 nM GST-Grp1-PH, followed by addition of 5 µL 40 nM TAMRA Red Fluorescent Probe in HNT buffer. Identity of the lipid group coupled to the TAMRA probe was not disclosed by Echelon Biosciences, but as Grp1 recognises specific lipid head groups in competition with PI(3,4,5)P3, it is likely to be a lipid with a similar head group such as inositol 1,3,4,5-tetrakisphosphate. After 1 hr equilibration, fluorescence polarisation was measured using a PHERAstar spectrofluorometer (BMG Labtech, Ortenberg, Germany) with the FP/540-20/590-20/590-20 optical module. The concentrations of WT protein complexes used in the assay were: 20–40 nM p85α/p110α, 80–160 nM p85α/p110β, and 80–160 nM p85α/p110δ for basal activities, and 1–2 nM p85α/p110α, 2–4 nM p85α/p110β, and 2–6 nM p85α/p110δ for pY2-stimulated activities. Concentrations of thawed protein aliquots were assessed as 280 nm absorbance by NanoDrop 1000 Spectrophotometer, and samples of preparations used in assays were resolved by SDS-PAGE to confirm equal amounts of each complex. Three technical replicates were used per assay on at least three different occasions. Relevant WT p85α/p110 catalytic isoforms were included in each assay as controls. Other controls were (1) Stop mix+probe without lipids, to assess maximum fluorescence polarisation. (2) Stop mix+probe with lipids, to determine the effect of lipids on maximum fluorescence polarisation. (3) Probe without stop mix or lipids to determine minimum fluorescence polarisation. (4) Probe with lipids but no stop mix to determine the effect of lipids on minimum fluorescence polarisation. Standard curves of diC8-PI(3,4,5)P3 in the presence of liposomes were included in every experiment and used to infer PIP3 generated by purified PI3K. GraphPad Prism v6.0 was used to generate sigmoidal standard curves by plotting log-transformed diC8-PI(3,4,5)P3 concentrations against fluorescence polarisation. Standard curves for each experiment provided the linear range within which PIP3 could be accurately determined. Preliminary studies found only trivial differences in measured fluorescence polarisation in the presence of pY2, which was thus omitted from subsequent standard curves. Quadruplicate interpolated PIP3 concentrations were averaged and normalised by reaction time (30 min) and enzyme concentration in pM to yield PI3K

activity in pmol PIP3/min/pM enzyme. Specific activities were then further normalised to activity of p85α WT controls in the presence of pY2 in each experiment.

## Generation of 3T3-L1 cells conditionally expressing p85α or p110α

3T3-L1 preadipocytes (Zenbio, lot 3T3062104, passage 8) were cultured, differentiated, and stained for neutral lipid with Oil Red O as previously described (*Huang-Doran et al., 2011*). Differentiation (visible as accumulation of multiple lipid droplets) served as a phenotypic assay validating cell identity. All cell lines used were confirmed mycoplasma free through a routine policy of monthly PCR-based screening. Generation of 3T3-L1 sublines inducibly expressing WT, Y657X, or R649W p85α was also previously reported (*Huang-Doran et al., 2016*; *Hussain et al., 2011*). The 3T3-L1 lines inducibly expressing p85α ΔEx11 or p110α H1047R were generated using essentially the same procedure, starting with In-Fusion subcloning of the p85a ΔEx11 cDNA insert from the pACEBac1 p85a ΔEx11 plasmid described above, and of PIK3CA H1047R cDNA (based on Uniprot P42336) derived by PCR of a pre-existing expression vector, into the pEN_Tmcs entry vector. After sequence verification these inserts were integrated into a pSLIK-Hygro lentiviral vector, packaged into VSV-G-pseudotyped lentiviral particles, using packaging plasmids pMDLg/pRRE and pRSVREV, and VSV-G envelope plasmid pVSV, and used to infect 3T3-L1 preadipocytes (plasmids listed in Key resources table). Transgene expression was induced with 1 µg/mL doxycycline for 72 hr, or with variable concentrations of doxycycline as indicated.

## Insulin signalling studies

3T3-L1 cells serum starved in DMEM containing 0.5% BSA for 16 hr were stimulated with 100 nM Actrapid insulin (Novo Nordisk) for 10 min. Monolayers were snap-frozen in liquid N2 and stored at –80°C. 200–600 µL co-immunoprecipitation (Co-IP) lysis (20 mM HEPES, 150 mM NaCl, 1.5 mM MgCl₂, 10% [vol/vol] glycerol, 1% [vol/vol] Triton X-100, 1 mM EGTA pH 7.4, 1 mM PMSF, 2 mM activated sodium orthovanadate, Complete Mini EDTA-free protease inhibitor cocktail, and PhosSTOP [Roche 04906837001], in Milli-Q Ultrapure water [Millipore]) or RIPA buffer (50 mM Tris-HCl pH 8.0, 150 mM NaCl, 1% NP-40, 0.5% sodium deoxycholate, 0.1% SDS with added Complete Mini EDTA-free protease inhibitor cocktail and PhosSTOP, in Milli-Q Ultrapure Water [Millipore]) was added to frozen cells before scraping of lysate into pre-chilled tubes, incubation on ice for 30 min and clearing by centrifugation. Protein was quantified using the Bio-Rad DC assay.

Lysates were mixed with NuPAGE LDS Loading buffer (Life Technologies, NP0008) supplemented with 5% β-mercaptoethanol and boiled before SDS-PAGE. For Co-IP lysates (150–300 µg protein) were incubated in 500 µL Co-IP buffer with immunoprecipitating antibody overnight at 4°C. 10 µg lysate mixed with NuPAGE SDS Loading buffer and 5% β-mercaptoethanol was stored to represent Co-IP input. Co-IP samples were incubated for 2 hr with 1.5 mg PBST-washed Protein G Dynabeads (Life Technologies, 10003D) at 4°C before centrifugation and bead removal using a DynaMag-2 Magnet (Invitrogen). Supernatants mixed 1:1 with Co-IP elution buffer (2X NuPAGE LDS, 100 mM NuPAGE Sample Reducing Agent [Invitrogen, NP0009], in Co-IP lysis buffer) were boiled for 10 min. Beads were washed five times in 50 µL Co-IP lysis buffer, before protein elution with 25 µL Co-IP elution buffer and boiling for 10 min. Input, supernatant, and Co-IP samples underwent SDS-PAGE using NuPAGE 4–12% gradient Bis-Tris gels (Thermo Fisher, NP0321BOX) in NuPAGE MOPS SDS Running Buffer (Life Technologies, NP0001). Transfer to nitrocellulose was performed using the iBlot Dry Blotting System (Invitrogen), with preincubation in Equilibration Transfer Buffer for proteins above 150 kDa.

For immunoblotting, blocked membranes were incubated overnight at 4°C in primary antibody, washed and incubated with HRP-linked anti-rabbit or anti-mouse IgG secondary antibody diluted 1:5000 in blocking buffer. Proteins were visualised using the ChemiDoc MP System (Bio-Rad) and band intensities quantified using Image Lab software (Bio-Rad).

## Statistical analysis

For quantitative data, all biological replicates (i.e. results of independent experiments on different days) are represented in figures, with paired points (e.g. with/without insulin in the same experiment) connected by lines. For fluorescence polarisation assays, each biological replicate shown is the mean of three technical replicates. Sample size for individual experiments was not pre-determined. To avoid the pitfalls of dichotomous significance testing on low-throughput biological datasets, we

used estimation statistics (Data Analysis using Bootstrap-Coupled ESTimation) with default settings (5000 resamples, BCa bootstrap) (*Ho et al., 2019*). This focuses on effect sizes and derives confidence intervals derived from bootstrapping for differences in means; the small (but typical) sample size of the experiments analysed limits reliable bootstrapping, but it offers additional confidence to the consistent patterns seen in independent replicates. We have indicated significance in main text figures with an asterisk, and show the 95% confidence intervals for mean differences in extended view figures.

## Acknowledgements

This work was supported by the Wellcome Trust through a grant to RKS (210752/Z/18/Z) and a studentship to PRT (102356/Z/13/Z). Additional support was from the UK Medical Research Council (MRC) (MC_UU_12012/5 and MC_U105184308 [to RLW]) and the Intramural Research Program of the National Institute of Allergy and Infectious Diseases, National Institutes of Health (to HCS). We thank Dr. Koneti Rao, Debra Long-Priel, and Angela Wang for clinical, technical, and regulatory assistance.

## Additional information

### Competing interests

Robert K Semple: Consulting for Novartis on clinical aspects of PIK3CA-related overgrowth, and for Alnylam, Amryt and AstraZeneca on clinical aspects of monogenic insulin resistance and lipodystrophy. The other authors declare that no competing interests exist.

### Funding

| Funder | Grant reference number | Author |
|---|---|---|
| Wellcome Trust | 10.35802/210752 | Robert K Semple |
| Wellcome Trust | 10.35802/102356 | Patsy R Tomlinson |
| Medical Research Council | MC_UU_12012/5 | Robert K Semple |
| Medical Research Council | MC_U105184308 | Roger L Williams |
| National Institute of Allergy and Infectious Diseases | Intramural | Helen Su |

The funders had no role in study design, data collection and interpretation, or the decision to submit the work for publication. For the purpose of Open Access, the authors have applied a CC BY public copyright license to any Author Accepted Manuscript version arising from this submission.

### Author contributions

Patsy R Tomlinson, Conceptualization, Data curation, Formal analysis, Investigation, Visualization, Methodology, Writing – original draft; Rachel G Knox, Formal analysis, Investigation, Visualization; Olga Perisic, Supervision, Investigation, Methodology, Writing – review and editing; Helen Su, Resources, Funding acquisition, Writing – review and editing; Gemma V Brierley, Supervision, Methodology, Writing – review and editing; Roger L Williams, Supervision, Funding acquisition, Investigation, Methodology, Writing – review and editing; Robert K Semple, Conceptualization, Resources, Data curation, Formal analysis, Supervision, Funding acquisition, Visualization, Writing – original draft, Project administration, Writing – review and editing

### Author ORCIDs

Roger L Williams  https://orcid.org/0000-0001-7754-4207
Robert K Semple  https://orcid.org/0000-0001-6539-3069

### Ethics

The APDS2 patient studied gave written informed consent for NIH IRB-approved research protocols 06-I-0015 and 09-I-0133 and was reported previously (Lucas et al., 2014b). Control wild-type (WT) and PIK3CA mutant dermal fibroblasts were obtained from healthy volunteers and patients with PIK3CA-related overgrowth spectrum (PROS) conditions (Lindhurst et al., 2012) also after obtaining

written evidence of informed consent, as part of studies approved by the UK National Research Ethics Committee (studies 18/EE/0068 and 12/EE/0405). Dermal fibroblasts were isolated from skin punch biopsies and cultured as previously described (Huang-Doran et al., 2016).

Reviewer #1 (Public review): https://doi.org/10.7554/eLife.94420.3.sa1
Reviewer #2 (Public review): https://doi.org/10.7554/eLife.94420.3.sa2
Author response https://doi.org/10.7554/eLife.94420.3.sa3

---

## Additional files

### Supplementary files
MDAR checklist

### Data availability
All data generated or analysed during this study are included in the manuscript and supporting files. All antibodies and plasmids used are listed in the Key resources table.

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
