## [Editor Report · eLife Assessment]

This **important** study reports on PIK3R1 mutations and a paradoxical mechanism of PIK3R1 signaling. The strength of evidence for the study is mostly **convincing**, as conclusions are supported by a variety of mutational strategies and cellular systems to look at interactions among signaling pathways.

---

## [Referee Report · Reviewer #1 (Public review)]

Summary:

This study provides convincing data showing that expression of the PIK3R1(deltaExon11) dominant negative mutation in Activated PI3K Delta Syndrome 1/2 (APDS1/2) patient-derived cells reduces AKT activation and p110δ protein levels. Using a 3T3-L1 model cell system, the authors show that overexpressed p85α(deltaExon 11) displays reduced association with the p110α catalytic subunit but strongly interacts with Irs1/2. Overexpression of PIK3R1 dominant negative mutants inhibit AKT phosphorylation and reduce cellular differentiation of preadipocytes. The experimental design, interpretation, and quantification broadly support the authors' conclusions, which establishes a new paradigm that warrants future studies.

Strengths:

The strength of this study is the clear results derived from Western blots analysis of cell signaling markers (e.g. pAKT1), and co-immunoprecipitation of PI3K holoenzyme complexes and associated regulatory factors (e.g. Irs1/2). The authors analyze a variety of PIK3R1 mutants (i.e. deltaExon11, E489K, R649W, and Y657X), which reveals a range of phenotypes that support the proposed model for dominant negative activity. The use of clonal cell lines with doxycycline induced expression of the PIK3R1 mutants (deltaExon 11, R649W, and Y657X) provides convincing experimental data concerning the relationship between p85α mutant expression and AKT phosphorylation in vivo. This approach for overexpression is excellent and should be utilized more broadly by cell biologists. The authors convincingly show that p85α (deltaExon11, R649W, or Y657X) is unable to associate with p110α but instead more strongly associates with Irs1/2 compared to wild type p85α. Overall, this article does a great job of motivating future studies of SHORT and APDS2 PIK3R1 mutants expressed from their endogenous loci (e.g. knock-in mice).

Weaknesses:

The limitations for this study lie in the complexity of the cell signaling pathway under investigation, rather than a lack of rigor by the authors. Future experimentation will help reconcile the cell type specific differences (e.g. APDS2 patient derived cells vs. the 3T3-L1 cell model system) in PIK3R1 mutant behavior reported by the authors. This is also intimately linked to variable expression of PIK3R1 mutants and cell-type specific regulatory factors. Although beyond the scope of this work, an unbiased proteomic study that broadly evaluates the cell signaling landscape could provide a more holistic understanding of the APDS2 and SHORT mutants compared to a candidate-based approach. Additional structural biochemistry of the p110α/p85α(deltaExon 11) complex is needed to explain why PIK3R1 mutant regulatory subunits do not strongly associate with the p110 catalytic subunit. A more comprehensive biochemical analysis of p110α/p85α, p110β/p85α, and p110δ/p85α mutant protein complexes will also be necessary to explain various cell signaling phenotypes. A minor limitation of this study is the use of single end point assays to measure PI3K lipid kinase activity in the presence of one regulatory input (i.e. RTK-derived pY peptide). An expanded biochemical analysis of purified mutant PI3K complexes across the canonical membrane signaling landscape will be important for deciphering how competition between wild-type and mutant regulatory subunits are regulated in different cell signaling contexts.

---

## [Referee Report · Reviewer #2 (Public review)]

Patsy R. Tomlinson et al; investigated the impact of different p85 alpha variants associated with SHORT syndrome or APDS2 on insulin mediated signaling in dermal fibroblasts and preadipocytes. They perform this study as APDS2 patients oftern present with features of SHORT syndrome. They found no evidence of hyperactive PI3K signalling monitored by pAKT in a APDS2 patient-derived dermal fibroblast cells. In these cells p110 alpha protein levels were comparable to levels in control cells, however, p110 delta protein levels were strongly reduced. Remarkably, the truncated APDS2-causal p85 alpha variant was less abundant in these cells than p85 alpha wildtype. Afterwards they studied the impact of ectopically expressed p85 alpha variants on insulin mediated PI3K signaling in 3T3-L1 preadipocytes. Interestingly they found that the truncated APDS2-causal p85 alpha variant impaired insulin induced signaling. Using immunoprecipitation of p110 alpha they did not find truncated APDS2-causal p85 alpha variant in p110 alpha precipitates. Furthermore, by immunoprecipitating IRS1 and IRS2 they observed that the truncated APDS2-causal p85 alpha variant was very abundant in IRS1 and IRS2 precipitates, even in the absence of insulin stimulation. These important findings add in an interesting way possible mechanistic explanation for the growing number of APDS2 patients described with features of SHORT syndrome.

Strengths:

Based on state-of-the-art functional studies, the authors show that the p85 alpha variant responsible for APDS2, known to be associated with increased PI3K-delta signaling, can attenuate PI3K-alpha signalling in preadipocytes, providing a possible mechanistic explanation for the growing number of APDS2 patients with features of SHORT syndrome.

Weaknesses:

The proposed paradigm is based on one cell line derived from an APDS2 patient and an overexpressing system. The investigation of a larger number of cell lines derived from APDS2 patients would further substantiate the conclusion.

---

## [Author Response]

The following is the authors’ response to the original reviews.

**eLife Assessment**
The authors identify new mechanisms that link a PIK3R1 mutant to cellular signaling and division in Activated PI3 Kinase Delta Syndrome 1 and 2 (APDS1/2). The conclusion that this mutant serves as a dominant negative form of the protein, impacting PI3K complex assembly and IRS/AKT signaling, is important, and the evidence from constitutive and inducible systems in cultured cells is convincing. Nevertheless, there are several limitations relating to differences between cell lines and expression systems, as well as more global characterization of the protein interaction landscape, which would further enhance the work.

We are pleased by this fair assessment, while noting that this work relates to APDS2 (PIK3R1-related) rather than APDS1 (PIK3CD-related). Our findings we believe are clear, but the observation that studies including more global proteomics/phosphoproteomics in cells expressing mutants at endogenous levels would add further insight is well made. We hope that this report may motivate such studies by laboratories with wider access to primary cells from patients and knock-in mice.

**Public Reviews**

**Reviewer #1 (Public Review):**
Summary:This study provides convincing data showing that expression of the PIK3R1(delta Exon11) dominant negative mutation in Activated PI3K Delta Syndrome 1/2 (APDS1/2) patient-derived cells reduces AKT activation and p110δ protein levels. Using a 3T3-L1 model cell system, the authors show that overexpressed p85α delta Exon 11 displays reduced association with the p110α catalytic subunit but strongly interacts with Irs1/2. Overexpression of PIK3R1 dominant negative mutants inhibits AKT phosphorylation and reduces cellular differentiation of preadipocytes. The strength of this article is the clear results derived from Western blots analysis of cell signaling markers (e.g. pAKT1), and co-immunoprecipitation of PI3K holoenzyme complexes and associated regulatory factors (e.g. Irs1/2). The experimental design, interpretation, and quantification broadly support the authors' conclusions.Strengths:The authors analyze a variety of PIK3R1 mutants (i.e. delta Exon11, E489K, R649W, and Y657X), which reveals a range of phenotypes that support the proposed model for dominant negative activity. The use of clonal cell lines with doxycycline-induced expression of the PIK3R1 mutants (ΔExon 11, R649W, and Y657X) provides convincing experimental data concerning the relationship between p85α mutant expression and AKT phosphorylation in vivo. The authors convincingly show that p85α delta Exon11, R649W, or Y657X is unable to associate with p110α but instead more strongly associates with Irs1/2 compared to wild type p85α. This helps explain why the authors were unable to purify the recombinant p110α/p85α delta Exon 11 heterodimeric complex from insect cells.Weaknesses:Future experimentation will be needed to reconcile the cell type specific differences (e.g. APDS2 patient-derived cells vs. the 3T3-L1 cell model system) in PIK3R1 mutant behavior reported by the authors.

This is a fair comment. It has been established for many years that relative protein levels even of wild type PIK3CA and PIK3R1 gene products influence sensitivity of PI3K to growth factor stimulation. Such issues of stoichiometry become exponentially more complicated when the numerous potential interactions among the full repertoire of Class 1 PI3K regulatory subunits (3 splice variants of PIK3R1, and also PIK3R2 and PIK3R3) and corresponding catalytic subunits (PIK3CA, PIK3CB, PIK3CD) are considered, and when different activities and stabilities of PIK3R1 mutants are added to the mix. It thus seems obvious to us that different levels of expression of different mutants in different cellular contexts will have different signalling consequences. We establish a paradigm in this paper using an overexpression system, and we strongly agree that this merits further investigation in a wider variety of primary cells (or cells with knock in at the endogenous locus), where available.

An unbiased proteomic study that broadly evaluates the cell signaling landscape could provide a more holistic understanding of the APDS2 and SHORT mutants compared to a candidate-based approach.

We agree. This would be highly informative, but we think would best be carried out in both “metabolic” and “immune” cells with endogenous levels of expression of SHORT or APDS2 PIK3R1 mutants. These are not all currently available to us, and require follow up studies.

Additional biochemical analysis of p110α/p85α delta Exon 11 complex is needed to explain why this mutant regulatory subunit does not strongly associate with the p110 catalytic subunit.

We agree. We present this observation in our overexpression system, which is clear and reproducible, even though somewhat surprising. The failure to bind p110a is likely not absolute, as sufficient p110a-p85a^ΔEx11^ was synthesised in vitro in a prior study to permit structural and biochemical studies, although a series of technical workarounds were required to generate enough heterodimeric PI3K to study in vitro given the manifest instability of the complex, particularly when concentrated (PMID 28167755). We already note in discussion that p85a can homodimerize and bind PTEN, likely among other partners, and it may be that the APDS2 deletion strongly favours binding to proteins that effectively compete with p110a. However this requires further study of the wider interactome of the mutant PIK3R1, which, as noted above, are beyond the scope of the current study.

It remains unclear why p85α delta Exon 11 expression reduces p110δ protein levels in APDS2 patient-derived dermal fibroblasts.

We caution that we only had the opportunity to study dermal fibroblasts cultured from a single APDS2 patient, as noted in the paper, and so replication of this finding in future will be of interest. Nevertheless the observation is robust and reproducible in these cells, and we agree that this apparently selective effect on p110d is not fully explained. Having said that, it has been observed previously that heterodimers of the ΔEx11 p85a variant with either p110a or p110d are unstable, and when the unstable complexes were eventually synthesised, p110a and p110d were demonstrated to show differences in engagement with the mutant p85, with greater disruption of inhibitory interactions observed for p110d (PMID 28167755). It is thus not a great stretch to imagine that as well as disinhibiting p110d more, the ΔEx11 p85a variant also destabilises the p85a-p110d complex more, potentially explaining its near disappearance in cells with low baseline p110d expression. Following on from the preceding question and response, however, is an alternative explanation, based on the 3T3-L1 overexpression studies in this paper, wherein we were unable to demonstrate binding of p110a by ΔEx11 p85a. If, in any given cellular context, the mutant p85 could bind p110d but not p110a, then the destabilising effect would be observed only for p110d. So in summary, we believe the selective effect on p110d is explained by differences in binding kinetics and heterodimer stability for different ΔEx11 p85a-containing complexes. The net effect of these differences may vary among cell types depending on relative levels of subunit expression.

This study would benefit from a more comprehensive biochemical analysis of the described p110α/p85α, p110β/p85α, and p110δ/p85α mutant protein complexes. The current limitation of this study to the use of a single endpoint assay to measure PI3K lipid kinase activity in the presence of a single regulatory input (i.e. RTK-derived pY peptide). A broader biochemical analysis of the mutant PI3K complexes across the canonical signaling landscape will be important for establishing how competition between wild-type and mutant regulatory subunits is regulated in different cell signaling pathways.

We agree that a wider analysis of upstream inputs and downstream network would be of interest, though as noted above the ultimate functional consequences of mutants will be an amalgam of any differential signalling effects of complexes that are stable enough to function, and differential effects of mutant p85a on the kinetics of distinct heterodimer assembly and stability. In this paper we seek to suggest a paradigm worthy of further, deeper assessment. We note that the search space here is large indeed (A. different cell types with differing profiles of PI3K subunit expression B. Multiple upstream stimuli and C. Multiple downstream outputs, with timecourse of responses an additional important factor to consider). These studies are realistically beyond the scope of the current work, but we hope that further studies, as suggested by the reviewer, follow.

**Reviewer #2 (Public Review)**
Summary:Patsy R. Tomlinson et al; investigated the impact of different p85alpha variants associated with SHORT syndrome or APDS2 on insulin-mediated signaling in dermal fibroblasts and preadipocytes. They find no evidence of hyperactive PI3K signalling monitored by pAKT in APDS2 patient-derived dermal fibroblast cells. In these cells p110alpha protein levels were comparable to levels in control cells, however, the p110delta protein levels were strongly reduced. Remarkably, the truncated APDS2-causal p85alpha variant was less abundant in these cells than p85alpha wildtype. Afterwards, they studied the impact of ectopically expressed p85alpha variants on insulin-mediated PI3K signaling in 3T3-L1 preadipocytes. Interestingly they found that the truncated APDS2-causal p85alpha variant impaired insulin-induced signaling. Using immunoprecipitation of p110alpha they did not find truncated APDS2-causal p85alpha variant in p110alpha precipitates. Furthermore, by immunoprecipitating IRS1 and IRS2, they observed that the truncated APDS2-causal p85alpha variant was very abundant in IRS1 and IRS2 precipitates, even in the absence of insulin stimulation. These important findings add in an interesting way possible mechanistic explanation for the growing number of APDS2 patients described with features of SHORT syndrome.Strengths:Based on state-of-the-art functional investigation the authors propose indicating a loss-of-function activity of the APDS2-disease causing p85alpha variant in preadipocytes providing a possible mechanistic explanation for the growing number of APDS2 patients described with features of SHORT syndrome.Weaknesses:Related to Figure 1: PIK3R1 expression not only by Western blotting but also by quantifying the RNA transcripts, e.g. mutant and wildtype transcripts, was not performed. RNA expression analysis would further strengthen the suggested impaired stabilization/binding.

It is not completely clear to us how further PIK3R1 mRNA analysis would enhance the points we seek to make. Perhaps the reviewer’s point is that changes in protein expression could be explained by reduced transcription rather than having anything to do with altered protein turnover? As shown in Figure 1 supplemental figure 1, sequencing cDNA from each of the primary cell lines studied indicates that both mutant and WT alleles are expressed at or close to 50% of the total mRNA for PIK3CA or PIK3R1 as relevant. While this is not strictly quantitative, allied to prior evidence that these are dominant alleles which require to be expressed to exert their effect, with no evidence for altered mRNA expression of these variants in prior studies, we don’t believe any further quantification of mRNA expression would add value.

Related to Figure 2As mentioned by the authors in the manuscript the expression of p110delta but also p110beta in 3T3-L1 preadipocytes ectopically expressing p85alpha variants has not been analyzed.

We agree that such determination would have been a useful addition to the study, but regretfully it was not undertaken in these modified 3T3-L1 cells at the time of study. However independent bulk RNAseq studies of the founder 3T3-L1 cells from which the stably transduced cells were generated, undertaken as part of an unrelated study, revealed the following relative levels of endogenous expression of PI3K subunit mRNA:

**Author response table 1. sa3table1:** 

ENSEMBL_ID	Gene	Description	logCPM	CPM
ENSMUSG00000041417	Pik3r1	p85/p55/p50alpha	7.65	201
ENSMUSG000000027665	Pik3ca	P110alpha	7.09	137
ENSMUSG00000032462	Pik3cb	P110beta	3.54	12
ENSMUSG00000039936	Pik3cd	P110delta	-0.14	1

We have not determined endogenous protein expression, and so have left the text of the discussion unchanged, simply noting that we have not formally assessed protein expression of p110d/p110b. However these transcriptomic findings suggest that p110d protein is likely either undetectable, or else present at extremely low levels compared to endogenous p110a. p110b also appears to be expressed at a much lower level than p110a. In our studies overexpressing mutant PIK3R1 and assessing insulin action, we believe we are largely or perhaps entirely assaying the effect of the mutants on p110a, in keeping with the fact that genetic and pharmacological studies have firmly established that it is p110a that is responsible for mediating the metabolic actions of insulin in adipose tissue and preadipocytes including 3T3-L1 (e.g. PMID 16647110). Indeed, to quote from this study, in 3T3-L1 “… inhibitors of p110b (TGX-115 and TGX-286) and p110d (IC87114 and PIK-23) had no effect on the insulin-stimulated phosphorylation of any protein in the PI3-K pathway.”

We have added the following sentence to the discussion:

“The current study has limitations. We have studied primary cells from only a single APDS2 patient, and in the 3T3-L1 cell model, we did not determine whether p110d protein could be detected. If not, this could explain the lack of detectable AKT phosphorylation with induction of Pik3r1 ΔEx11. Indeed, previous pharmacological studies in 3T3-L1 adipocytes has shown that selective inhibition of p110d or p110b does not alter insulin-induced phosphorylation of any protein studied in the PI3-K pathway, attesting to the dominance of p110a in insulin action in this cell model (Knight et al, 2006).”

Furthermore, a direct comparison of the truncated APDS2-causal p85alpha variant with SHORT syndrome-causal p85alpha variants in regard to pAKT level, and p85alpha expression level has not been performed.These investigations would further strengthen the data.

The cell lines conditionally expressing SHORT syndrome variants have been reported already, as cited (PMID: 27766312). Remarkably, the degree of inhibition of insulin-stimulated signalling is actually less pronounced for the SHORT syndrome variants than for the overexpressed APDS2 variant, as seen in the excerpt from the prior paper below. In this prior paper the maximum insulin concentration used, 100nM, was the concentration used in the current study. While overexpression of the APDS2 p85a variant ablated the response to insulin entirely, it is still seen in the prior study, albeit at a clearly reduced level.

Related to Figure 3The E489K and Y657X p85alpha variants should be also tested in combination with p110delta in the PI3K activity in vitro assay. This would help to further decipher the overall impact, especially of the E489K variant.

We agree that this would make our data more complete, but for logistical reasons (primarily available personnel) we were compelled to constrain the number of p85-p110 combinations we studied. We elected to prioritise the PIK3R1 R649W variant as by far the most common causal SHORT syndrome variant, and as the variant showing the “cleanest” functional perturbation, namely severely impaired or absent ability to dock to phosphotyrosines in cognate proteins. The paradox that we sought to explain in this paper, namely the phenotypic combination of gain-of-function APDS2 with loss-of-function SHORT syndrome features holds only for APDS2 PIK3R1 variants, and so while it is interesting to document that the canonical SHORT syndrome variant also inhibits PI3Kb and PI3Kd activation in vitro, this was not the main purpose of our study.

**Reviewer #1 (Recommendations For The Authors):**
Points of clarification and suggestions for improving the manuscript:(1) Explain whether there are any PIK3R1-independent genetic alterations in the APDS2 and PROS-derived cell lines. For example, are there differences in the karyotype of mutant cell lines compared to wild-type cells?

Karyotypic abnormalities are not an established feature of either PROS or APDS2, and the patients from whom cells were derived were documented to be of normal karyotype. Karyotypic abnormalities acquired during cell culture would not be unprecedented, but confirming normal karyotypes in primary cell lines where there is no specific reason to suppose any alteration exceeds normal expectations for primary cell studies, and so this has not been undertaken.

(2) When introducing the APDS2-associated PIK3R1 mutation (lines 126-128), the authors describe both the exon 11 skipping and in-frame deletions. I recommend rewording this sentence to say exon 11 skipping results in an in-frame deletion of PIK3R1. The current wording makes it seem like APDS2-derived cells contain two genetic perturbations: (1) exon 11 skipping and (2) in-frame deletion. Include a diagram in Figure 1 to help explain the location of the mutations being studied in relationship to the PIK3R1 gene sequence and domains (i.e. nSH2, iSH2, cSH2). The description of the exon 11 skipping and in-frame deletions (lines 126-128) would benefit from having a complementary figure that diagrams the location of these mutations in the PIK3R1 gene.

On review we agree that clarity of description could be enhanced. We have now edited these lines as follows:

“We began by assessing dermal fibroblasts cultured from a previously described woman with APDS2 due to the common causal PIK3R1 mutation. This affects a splice donor site and causes skipping of exon 11, leading to an in-frame deletion of 42 amino acids (434-475 inclusive) in the inter-SH2 domain, which is shared by all PIK3R1 isoforms (Patient A.1 in (Lucas et al., 2014b))(Figure 1 figure supplement 1).”

We have moreover introduced a further figure element including a schematic of all PIK3R1 mutations reported in the current study (new Figure 1 figure supplement 1)

(3) For Figure 2, I recommend including a cartoon that illustrates the experimental design showing the induced expression of PIK3R1 mutants, R649W and Y657X, in the background of the wild-type endogenous gene expression.

Such a figure element has now been generated and included as Figure 2 figure supplement 1, duly called out in the results section where appropriate.

(4) For the data plotted in Figure 1B-1C, please clarify whether the experiments represent a single patient or all 3-4 patients shown in Figure 1A.

Each datapoint shown represents one of the patients in the immunoblots, with all patients included. Each point in turn is the mean from 3 independent experiments. We have added the following to the Figure legend:

“(B)-(E) quantification of immunoblot bands from 3 independent experiments shown for phosphoAKT-S473, phosphoAKT-T308, p110d and p110a respectively. Each point represents data from one of the patient cell lines in the immunoblots. Paired datapoints +/- insulin are shown in (B) and (C), and dotted lines mark means.”

(5) I recommend rewording the following sentence: "Given this evidence that APDS2-associated PIK3R1 delta Exon 11 potently inhibits PI3Kα when overexpressed in 3T3-L1 preadipocytes," to say "... potently inhibits PI3Kα signaling when overexpressed in 3T3-L1 preadipocytes." The data shown in Figures 1 and 2 do not support a direct biochemical inhibition of PI3Kα lipid kinase activity by p85α (delta Exon 11).

This edit has been made.

(6) Provide more discussion concerning the percentage of humans with APDS2 or SHORT syndrome that contain the mutations discussed in this paper. How strong is the genotype-phenotype link for these diseases? Are these diseases inherited or acquired through environmental stresses?

Both APDS2 and SHORT syndrome are very well established, highly penetrant and stereotyped monogenic disease. APDS is defined by the presence of activating PIK3R1 mutations such as the one studied here (by far the commonest causal mutation). SHORT syndrome clinically has some superficial resemblance to other human genetic syndrome including short stature, but when careful attention is paid to characteristic features it is nearly universally attributable to loss-of-function PIK3R1 mutations with the single exception of one case in which a putatively pathogenic PKCE mutation was described (PMID: 28934384). Although both syndromes are monogenic it is often not accurate to refer to them as inherited, as, particularly in SHORT syndrome, de novo mutations (i.e. not found in either parent) are common. Environmental modifiers of phenotypes have not been described. To the introduction has now been added the comment that both conditions are highly penetrant and monogenic.

(7) The data presented in Figure 5 would benefit from additional discussion and citations that describe the molecular basis of the interaction between PI3K and Irs1/2. What studies have previously established this is a direct protein-protein interactions? Are there PI3K mutants that don't interact with Irs1/2 that can be included as a negative control? Alternatively, the authors can simply reference other papers to support the mechanism of interaction.

There is a voluminous literature dating back to the early 1990s documenting the mode of interaction of PI3K with Irs1/2. Relevant papers have now been cited as requested:

p85-Irs1 binding: PMID 1332046 (White lab, PNAS 1992)

p85-Irs2 binding: PMID 7675087 (White lab, Nature 1995)

“This may be important, as p85a mediates recruitment of PI3K to activated tyrosine kinase receptors and their tyrosine phosphorylated substrates, including the insulin-receptor substrate proteins Irs1 (PMID 1332046) and Irs2 (PMID 7675087).”

Regarding PI3K mutants that don't interact with Irs1/2, the SHORT syndrome mutant R649W which we include in this study is perhaps the best example of this, so it is both disease-causing and functions as such a negative control.

(8) To see the effect of the dominant negative delta Exon 11, the truncated p85α needs to be super stoichiometric to the full-length p85α (Figure 2 - Supplemental Figure 2). This is distinct from the results in Figure 1 showing the ADPS2-derived dermal fibroblast express 5-10x lower levels of p85α delta Exon 11 compared to full-length p85α (Figure 1A), but still strongly inhibits pAKT S473 and T308 (Figure 1B-1C). The manuscript would benefit from more discussion concerning the cell type specific differences in phenotypes. Alternatively, do the APDS2-derived dermal fibroblasts have other genetic perturbations that are not accounted for that potentially modulate cell signaling differently compared to 3T3-L1 preadipocytes?

The reviewer is astute to point out this apparent contrast. First of all, we have no reason to suppose there is any specific, PI3K-modifying genetic perturbation present in the primary dermal fibroblasts studied, although of course the genetic background of these cells is very distinct to that of 3T3-L1 mouse embryo fibroblasts. Related to such background differences, however, substantial variability is usually apparent in insulin-responsiveness even of healthy control dermal fibroblasts. This means that caution should be exercised in extrapolating from studies of the primary cells of a single individual. To illustrate this, we point the reviewer to our 2016 study in which we extensively studied the dermal fibroblasts of a proband with SHORT syndrome due to PIK3R1 Y657X:

From this study we conclude that A. WT controls show quite substantial variation in insulin-stimulated AKT phosphorylation and B. even the SHORT syndrome p85a Y657X variant, expressed at higher levels that WT p85a in dermal fibroblasts, does not produce an obvious decrease in insulin-stimulated AKT phosphorylation, despite extensive evidence from other human cell studies and knock-in mice that it does indeed impaired insulin action in metabolic tissues. For both these reasons we are not convinced that the lower insulin-induced AKT phosphorylation we described in Figure 1 should be overinterpreted until reproduced in other studies with primary cells from further APDS2 patients. This is why we did not comment more extensively on this. We now add the following qualifier in results:

“Despite this, no increase in basal or insulin-stimulated AKT phosphorylation was seen in APDS2 cells compared to cells from wild-type volunteers or from people with PROS and activating PIK3CA mutations H1047L or H1047R (Fig 1A-C, Fig 1 figure supplement 3A,B). Although insulin-induced AKT phosphorylation was lower in fibroblasts from the one APDS2 patient studied compared to controls, we have previously reported extensive variability in insulin-responsiveness of primary dermal fibroblasts from WT controls. Moreover even primary cells from a patient expressing high levels of the SHORT syndrome-associated p85a Y657X did not show attenuated insulin action, so we do not believe the reduced insulin action in APDS2 cells in the current study should be overinterpreted until reproduced in further APDS2 cells.”

Nevertheless we remind the reviewer that the main purpose of our primary cell experiment was to determine if there were any INCREASE in basal PI3K activity, or any difference in p110a or p110d protein levels, and we regard our findings in these regards to be clear.

The manuscript would benefit from additional explanation concerning why the E489K, R649W, and Y657X are equivalent substitutes for the characterization of p110α/p85α delta Exon 11. Perhaps a more explicit description of these mutations in relationship to the location of p85α delta Exon 11 mutation would help. I recommend including a diagram in Figure 3 showing the position of the delta Exon 11, E489K, R649W, and Y657X mutations in the PIK3R1 coding sequence. B. Also, please clarify whether all three holoenzyme complexes were biochemically unstable (i.e. p110α/p85α, p110β/p85α, p110δ/p85α) when p85α delta Exon 11 was expressed in insect cells.

A. Whether or not E489K, R649W and Y657X are “equivalent” to the APDS2 mutant is not really a meaningful issue here. These mutants are being studied because they cause SHORT syndrome without immunodeficiency, while the APDS2 mutant causes APDS2 often with features of SHORT syndrome. That is, it is naturally occurring mutations and the associated genotype-phenotype correlation that we seek to understand. Of the 3 SHORT syndrome causal mutations chosen, R649W is by far the commonest, effectively preventing phosphotyrosine binding, Y657X has the interesting attribute that it can be discriminated from full length p85 on immunoblots due to its truncation, and is moreover a variant that we have studied in cells and mice before, while the rarer E489K is an interesting SHORT syndrome variant as it is positioned more proximally in the p85a protein than most SHORT syndrome causal variants. All variants studied are now illustrated in the new Figure 1 figure supplement 1. B. Regarding stability of PI3K heterodimers containing the APDS2 p85a variant, we tried extensively to purify p110a and p110d complexes without success despite several approaches to optimise production. We did not try to synthesise the p110b-containing complex.

(10) I recommend presenting the results in Figure 4 before Figure 3 because it provides a good rationale for why it's difficult to purify the p110α/p85α delta Exon 11 holoenzyme from insect cells.

This would be true of p110d were studied in Figure 4 but it is not. Figure 4 looks instead at effects on p110a of heterologous overexpression of mutant p85, is a natural lead in to the ensuing figures 5 and 6, and we do not agree it would add value or enhance flow to swap Figures 3 and 4.

(11) The authors show that overexpression of the p85α delta Exon 11 did not result in p110α/p85α delta Exon 11 complex formation based on co-immunoprecipitation. Do the authors get the same result when they co-immunoprecipitation p110α/p85α and p110δ/p85α in the APDS2-derived dermal fibroblasts used in Figure 1A?

This is an interesting question but not an experiment we have done. It is not unfeasible, but generating enough cells to undertake IP experiments of this nature in dermal fibroblasts is a significant undertaking, and with finite resources available and only one primary cell line to study we elected not to pursue this.

Details in Methods section:(1) Include catalog numbers and vendors for reagents (e.g. lipids, PhosSTOP, G-Dynabeads, etc.). There is not enough information provided to reproduce this work.

We have now added all vendors and catalogue numbers where relevant.

(2) Concerning the stated lipid composition (5/10/15/45/20/5 %) in the liposome preparation protocol. Please clarify whether these numbers represent molar percentages or mg/mL percentages.

We have now added that this is expressed as “(wt/vol)”

(3) What is the amino acid sequence of the PDGFR (pY2) peptide used for the PI3K activity assay?

This assay has been published and references with detailed methods are cited. For clarity, however we now say:

“PI(3,4,5)P3 production was measured by modified PI3-Kinase activity fluorescence polarisation assay (Echelon Biosciences, Salt Lake City, UT, USA). 10μL reactions in 384-well black microtitre plates used 1mM liposomes containing 50μM PI(4,5)P2, optimised concentrations of purified PI3K proteins, 100μM ATP, 2mM MgCl2, with or without 1μM tyrosine bisphosphorylated 33-mer peptide derived from mouse PDGFRβ residues 735-767, including phosphotyrosine at positions 740 and 751 (“pY2”; 735-ESDGGYMDMSKDESIDYVPMLDMKGDIKYADIE-767; Cambridge peptides).”

(4) Include a Supplemental file containing a comprehensive description of the plasmids and coding sequencing used in this study.

Such a supplemental file has been created and is included as Table 2

Minor points of clarification, citations, and typos:(1) Clarify why Activated PI3K Delta Syndrome 1 (APDS1) is thus named APDS2. See lines 71-72 of the introduction. Also see line 89: "...is common in APDS2, but not in APDS1." Briefly describe the difference between APDS1 and APDS2?

This is described in the introduction, but we apologise if our wording was not sufficiently clear. We have tried now to remove any ambiguity:

“Some PIK3R1 mutations reduce basal inhibition of catalytic subunits, usually due to disruption of the inhibitory inter-SH2 domain, and are found in cancers (Philp et al, 2001) and vascular malformations with overgrowth (Cottrell et al, 2021). In both diseases, hyperactivated PI3Ka, composed of heterodimers of PIK3R1 products and p110a, drives pathological growth. Distinct inter-SH2 domain PIK3R1 mutations, mostly causing skipping of exon 11 and deletion of residues 434-475, hyperactivate PI3Kd in immune cells, causing highly penetrant monogenic immunodeficiency (Deau et al, 2014; Lucas et al, 2014b). This phenocopies the immunodeficiency caused by genetic activation of p110d itself, which is named Activated PI3K Delta Syndrome 1 (APDS1) (Angulo et al, 2013; Lucas et al, 2014a). The PIK3R1-related syndrome, discovered shortly afterwards, is thus named APDS2.”

(2) Figure legend 1. Clarify reference to "Figure EV2".(3) Figure legend 2. Clarify reference to "Figure EV3".(4) Figure legend 3. Clarify reference to "Figure EV5".

Thank you for pointing out this oversight, arising from failure to update nomenclature fully between versions. “EV” figures actually are the figure supplements in the submission. All labels have now been updated.

(5) For Figure 1 - supplemental figure 1C, indicate experimental conditions on the blot (e.g. -/+ insulin).

This is now added

(6) Figure 4B, y-axis. Clarify how data was quantified. Perhaps reword "(% WT without DOX)" for clarity.

We have left the Y axis label as it is, but have added the following to the figure legend:

“(B) Quantification of immunoblot bands from immunoprecipitates from 3 independent experiments, expressed as a percentage relative to the intensity of the band in WT cells without doxycycline exposure.”

(7) In the results section (lines 117-124), please explicitly state whether the described mutations are homo- or heterozygous.

All mutations are heterozygous, as now explicitly stated

(8) I recommend spelling out the SHORT and APDS2 acronyms in the abstract to make this study more accessible.

We respectfully disagree that such spelling out in the abstract would improve accessibility. Both acronyms are clunky and wordy and are more likely to obscure meaning by squeezing out other words in the abstract. APDS is already spelled out in the introduction, and we now add the following for SHORT syndrome:

“More surprisingly, phenotypic overlap is reported between APDS2 and SHORT syndrome. SHORT syndrome, named for the characteristic developmental features (Short stature, Hyperextensibility, Hernia, Ocular depression, Rieger anomaly, and Teething delay) is caused by loss of PI3Ka function due to disruption of the phosphotyrosine-binding C-terminal SH2 domain (Chudasama et al, 2013; Dyment et al, 2013; Thauvin-Robinet et al, 2013).”

(9) I recommend explaining in more detail or rewording the following jargon/terms to make the writing more accessible to a broad audience: "reduced linear growth" (line 83) and "larger series" (line 86). I assume "reduced linear growth" is height.

Edited as follows:

“It features short stature, insulin resistance, and dysmorphic features (Avila et al, 2016). In recent years, both individual case reports (Bravo Garcia-Morato et al, 2017; Petrovski et al, 2016; Ramirez et al, 2020; Szczawinska-Poplonyk et al, 2022) and larger case series (Elkaim et al, 2016; Jamee et al, 2020; Maccari et al, 2023; Nguyen et al, 2023; Olbrich et al, 2016; Petrovski et al., 2016) have established that many people with APDS2 have overt features of SHORT syndrome, while, more generally, linear growth impairment is common in APDS2, but not in APDS1.”

(10) For clarity, reword lines 214-215 to read, "No increase in p110α levels was seen on conditional overexpression of wild-type or R649W p85α."

Change made, thank you

(11) Figure 6A - Western blot label says, "657X" instead of "Y657X."

Now corrected

(12) Lines 214-215: For clarity, reword the sentence to say, "No increase in p110α was seen on conditional overexpression...".

REPEAT OF POINT 10 ABOVE

(13) Clarify what interactions are being competed for in the following statement: "... delta Ex11 may exert its inhibitory action by competing with PI3K holoenzyme" (lines 237-238). Are you referring to the interaction between p110α and p85α or the interaction between p110α/p85α and another protein?

We have endeavoured to clarify by editing as follows:

“As APDS2 p85a ΔEx11 does not appear to displace wild-type p85a from p110a despite strong overexpression, it is likely that there are high levels of truncated p85a unbound to p110a in the cell. This may be important, as p85a mediates recruitment of PI3K to activated tyrosine kinase receptors and their tyrosine phosphorylated substrates, including the insulin-receptor substrate proteins Irs1 and Irs2. Excess free regulatory subunits compete with heterodimeric PI3K holoenzyme for binding to these phosphotyrosines (Ueki et al., 2002), raising the possibility that excess free, truncated APDS2 p85a ΔEx11 may exert its inhibitory action similarly by outcompeting PI3K holoenzyme for phosphotyrosine binding.”

(14) Provide more information about the following statement and how it relates to the mutations in this study: "Homozygous truncating PIK3R1 mutations abolishing p85α expression while preserving p55α and p50α produce agammaglobulinaemia" (lines 271-272). The manuscript would benefit from a more explicit description of the nature of these mutations.

This wording seems to us to be explicit, however we agree that a schematic of PIK3R1 genotype-phenotype correlation, as requested elsewhere, would help readers. Such a schematic is now included as Figure 1 figure supplement 1.

(15) Typo on line 299: "unclike".

Corrected.

(16) The data presented in this study support a model in which p85α (ΔExon 11) expression functions as a dominant negative. Please clarify why in the discussion section you explain that p85α (ΔExon 11) activates PI3K. For example, "...skipping of exon 11, were shown in 2014 to activate PI3K..." (lines 290-291), "...activate PI3Kδ on one hand..." (line 309); "...APDS2 mutations in PIK3R1 has mixed consequences, producing greater hyperactivation of p110δ than p110α" (lines 354-355).

We do not entirely understand the reviewer’s question and thus request here. p85α (ΔExon 11) activates PI3Kd in immune cells and in vitro, and this is accepted, based on numerous reports, to be the mechanism underlying immunodeficiency. We do not challenge this, and cite evidence for any such claims in our report. The dominant negative activity we describe here towards PI3Ka activation is based not on inhibition of mutant-containing heterodimer, but rather on destabilisation of and/or competition with heterodimeric WT holoenzyme. This is the basis of the model we present; that is, a finely balanced competition between enzymic activation and mutant holoenzyme destabilisation and competition of mutant free p85a with WT holoenzyme, whose net effect likely differs among cells and tissues, most likely based on the repertoire and proportions of PI3K subunit expression. If the reviewer has specific suggestions for us that will make this point clearer still we should be happy to consider them.

(17) Provide references for the statements in lines 349-353 of the discussion.

This brief closing paragraph is a succinct recap and summary of the key points made throughout the manuscript and thoroughly referenced therein. We prefer to keep this section clean to maximise clarity, but are happy to copy references from the various other places in the manuscript to back up these assertions if this is preferred by the editorial team. Current text:

“In summary, it is already established that: A. genetic activation of PIK3CD causes immunodeficiency without disordered growth, while B. inhibition of PIK3R1 recruitment to RTKs and their substrates impairs growth and insulin action, without immunodeficiency, despite all catalytic subunits being affected and C. loss of p85α alone causes immunodeficiency.”

**Reviewer #2 (Recommendations For The Authors):**
In the abstract line 42 I would rather talk from SHORT syndrome like features.

Some patients do indeed meet the criteria for SHORT syndrome, but there is a spectrum. We have thus added this qualification and removed “short stature” to maintain the word count, as this is itself a SHORT syndrome-like feature.

Line 74 It would be helpful for the reader to give the amino-acid exchange and affected position of this single case.

We agree. Now added.

Furthermore, an illustration indicating the location of the different PIK3R1 variants on the p85 alpha level would be helpful for the reader.

As noted above such a figure element is now included as Figure 1 figure supplement 1 and duly called out in the text

The sentence in lines 298-300 makes no sense to me. Do you mean, unlike APDS1 murine models?

We agree, on review, that this paragraph is convoluted and makes a simple observation complex. We have rewritten now in what we hope is a more accessible style:

“Thus, study of distinct PIK3R1-related syndromes shows that established loss-of-function PIK3R1 mutations produce phenotypes attributable selectively to impaired PI3Ka hypofunction, while activating mutations produce phenotypes attributable to selectively increased PI3Kd signalling. Indeed, not only do such activating mutations not produce phenotypes attributable to PI3Ka activation, but they surprisingly have features characteristic of impaired PI3Ka function.”

Line 321 I propose including the notion of different cells: “The balance between expression and signalling in different cells may be a fine one ...”

This change has been made

Line 352 C. loss replace with complete loss.

“C.” actually denotes the last in a list after “A.” and “B.”. We have now used bold to emphasise this, but we imagine house style may dictate how we approach this.